# Host receptor-targeted therapeutic approach to counter pathogenic New World mammarenavirus infections

Brady T. Hickerson[1,11,12], Tracy R. Daniels-Wells [2,12], Cristian Payes[3], Lars E. Clark [4], Pierre V. Candelaria[2], Kevin W. Bailey[1], Eric J. Sefing[1], Samantha Zink [5], James Ziegenbein[5], Jonathan Abraham [4,6], Gustavo Helguera [3,5 ✉], Manuel L. Penichet[2,7,8,9,10 ✉] & Brian B. Gowen [1 ✉]

Five New World mammarenaviruses (NWMs) cause life-threatening hemorrhagic fever (HF). Cellular entry by these viruses is mediated by human transferrin receptor 1 (hTfR1). Here, we demonstrate that an antibody (ch128.1/IgG1) which binds the apical domain of hTfR1, potently inhibits infection of attenuated and pathogenic NWMs in vitro. Computational docking of the antibody Fab crystal structure onto the known structure of hTfR1 shows an overlapping receptor-binding region shared by the Fab and the viral envelope glycoprotein GP1 subunit that binds hTfR1, and we demonstrate competitive inhibition of NWM GP1 binding by ch128.1/IgG1 as the principal mechanism of action. Importantly, ch128.1/IgG1 protects hTfR1-expressing transgenic mice against lethal NWM challenge. Additionally, the antibody is well-tolerated and only partially reduces ferritin uptake. Our findings provide the basis for the development of a novel, host receptor-targeted antibody therapeutic broadly applicable to the treatment of HF of NWM etiology.

[1] Department of Animal, Dairy and Veterinary Sciences, Utah State University, Logan, UT, USA. [2] Division of Surgical Oncology, Department of Surgery, David Geffen School of Medicine at UCLA, Los Angeles, CA, USA. [3] Instituto de Biología y Medicina Experimental (IBYME CONICET), Buenos Aires, Argentina. [4] Department of Microbiology, Blavatnik Institute, Harvard Medical School, Boston, MA, USA. [5] Department of Chemistry and Biochemistry, University of California, Los Angeles (UCLA), Los Angeles, CA, USA. [6] Department of Medicine, Division of Infectious Diseases, Brigham and Women's Hospital, Boston, MA, USA. [7] Department of Microbiology, Immunology and Molecular Genetics, David Geffen School of Medicine at UCLA, Los Angeles, CA, USA. [8] UCLA Molecular Biology Institute, Los Angeles, CA, USA. [9] UCLA Jonsson Comprehensive Cancer Center, Los Angeles, CA, USA. [10] UCLA AIDS Institute, Los Angeles, CA, USA. [11] Present address: Division of Biotechnology Review and Research-III, Office of Biotechnology Products, Center for Drug Evaluation and Research, Food and Drug Administration, Silver Spring, MD, USA. [12] These authors contributed equally: Brady T. Hickerson, Tracy R. Daniels-Wells. ✉email: gustavoh@ibyme.conicet.gov.ar; penichet@mednet.ucla.edu; brian.gowen@usu.edu

Mammarenaviruses (family *Arenaviridae*) are enveloped, single-stranded, bisegmented, mostly rodent-borne RNA viruses, which are classified into Old World and New World lineages[1]. The New World mammarenaviruses (NWMs) are further divided into clades A, B, C, and D[2]. Several mammarenaviruses can cause severe hemorrhagic fever (HF) in humans. In the case of the NWMs, all the human pathogens belong to clade B and include Junín, Machupo, Guanarito, Sabiá, and Chapare viruses, which cause Argentine, Bolivian, Venezuelan, Brazilian, and Chapare HF, respectively. Transmission of these viruses to humans most commonly occurs through inhalation of aerosolized viral particles or direct contact of a wound with virus-containing rodent excreta or secreta[2]. Case fatality rates for untreated cases of NWM HF can be as high as 35% and countermeasures are available only for Argentine HF in the form of shrinking supplies of convalescent plasma derived from survivors and the live-attenuated Junín virus (JUNV) vaccine, Candid#1[3,4]. Accordingly, these agents are considered priority pathogens by federal and international public health agencies[5–7].

A distinguishing feature of the pathogenic NWMs is the ability to enter cells through human transferrin receptor 1 (hTfR1)[8–10], a ubiquitously expressed protein that is involved in the cellular uptake of iron[11,12]. The ability to use hTfR1 also appears to be a major determinant that defines whether a NWM can cause severe disease, as nonpathogenic NWMs can enter human cells in a hTfR1-independent manner without causing disease[2]. The binding site of the Machupo virus (MACV) envelope glycoprotein GP1 attachment subunit has been mapped to a common region of the apical domain of hTfR1[9]. This region is outside of the binding sites of the two major physiological ligands of TfR1, transferrin (Tf), and hereditary hemochromatosis protein (HFE), and is only known to interact with ferritin[12–15]. Structural studies have elucidated the atomic nature of the GP1 interaction with hTfR1 and suggest that pathogenic NWMs acquired the capacity to infect humans accidentally by co-evolution with the TfR1 of their respective reservoir host[16].

The conserved binding of pathogenic NWMs to a single epitope in the apical domain of hTfR1 presents a potential target for the development of broadly active therapeutics that disrupt viral GP1 attachment to hTfR1 without interfering with cellular uptake of iron. Indeed, we have previously shown that ch128.1/IgG3, an antibody targeting the apical region of hTfR1, can block the internalization of pseudoviruses displaying envelope glycoproteins of all known pathogenic NWMs[10]. The ch128.1/IgG3 antibody also inhibits cellular entry by a related North American NWM member of the Whitewater Arroyo virus species complex that utilizes hTfR1[17]. While promising, further development of this potential therapeutic approach has been hindered by the lack

of an accessible and cost-effective small-animal model compatible with the antigenic specificity of the ch128.1/IgG3 antibody, which specifically recognizes both human and Old World non-human primate TfR1[10].

Adult mice are refractory to severe infection and disease following pathogenic NWM challenge[18,19]; however, we have recently found that immunocompetent 3-week-old transgenic mice expressing hTfR1 are highly susceptible to JUNV infection and develop lethal disease[20]. The development of this novel small-animal model provides a valuable tool for the study of JUNV pathogenesis and a platform for the evaluation of therapeutic interventions that interfere with virus binding to hTfR1 in vivo. In the present report, we demonstrate the therapeutic potential of an IgG1 version of ch128.1 (ch128.1/IgG1) and its variant with impaired FcγR and C1q binding (ch128.1/IgG1 mutant) to not only inhibit JUNV infection in vitro but also in vivo using the novel hTfR1 mouse JUNV infection model. We also report the crystal structure of the Fab fragment of the ch128.1/IgG1 antibody (Fab ch128.1/IgG1) and computationally evaluate its interaction with hTfR1 using biochemical assays to confirm competitive inhibition between the antibody and MACV GP1. Further, we show that the antibody has low or no toxicity in human hematopoietic progenitor cells and only minimally competes with the binding and uptake of ferritin. Our findings serve as a proof-of-concept that administration of antibodies targeting the apical domain of hTfR1 protect against lethal disease and collectively support the continued development of this novel, host receptor-targeted therapeutic approach.

## Results

**Inhibition of JUNV infection by ch128.1/IgG1 and ch128.1/IgG1 mutant in cell culture**. The ch128.1/IgG3 antibody has previously been shown to inhibit JUNV infection in cell culture[10]. Prior to initiating mouse studies, the ch128.1/IgG1 and a ch128.1/IgG1 mutant antibody modified to inactivate Fc-mediated effector functions were evaluated for antiviral activity against the Candid#1 JUNV vaccine (JUNV-C1) and pathogenic JUNV Romero strains in cell culture (Fig. 1). The IgG1 isotype was selected with the long-term goal of advancing the development of ch128.1 antibody as a human therapeutic. Consistent with previous results with ch128.1/IgG3 inhibiting infection with an attenuated strain of JUNV at a concentration of 200 nM[10], potent inhibition was observed with both ch128.1/IgG1 and the mutant antibody. The 90% effective concentration (EC$_{90}$) of ch128.1/IgG1 and ch128.1/IgG1 mutant against JUNV-C1 were 0.73 nM and 0.46 nM, with selectivity index (SI) values of >1370 and >2174, respectively. The inhibitory activity of the antibodies

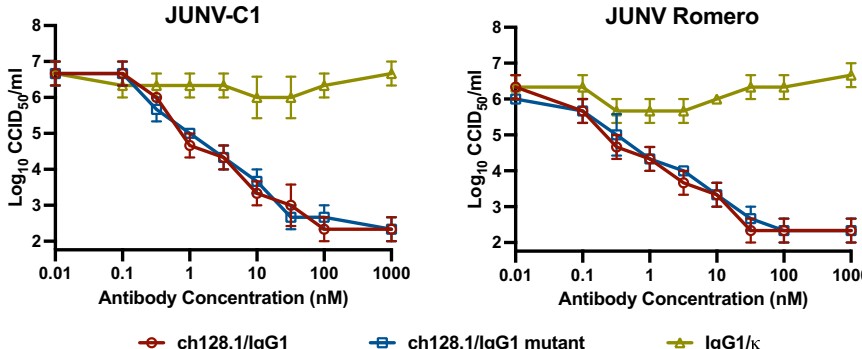

**Fig. 1 In vitro inhibition of JUNV infection by anti-hTfR1 antibodies.** Virus titers determined by endpoint titration of culture supernatant collected 4 days after infection from A549 cell cultures infected with JUNV-C1 or JUNV Romero and treated with increasing concentrations of ch128.1/IgG1, ch128.1/IgG1 mutant, or control IgG1/κ antibodies. Means ± standard deviations (SDs) of experimental replicates ($n = 3$) are shown. Fifty percent cell culture infectious dose (CCID$_{50}$). Source data are included in the Source Data file.

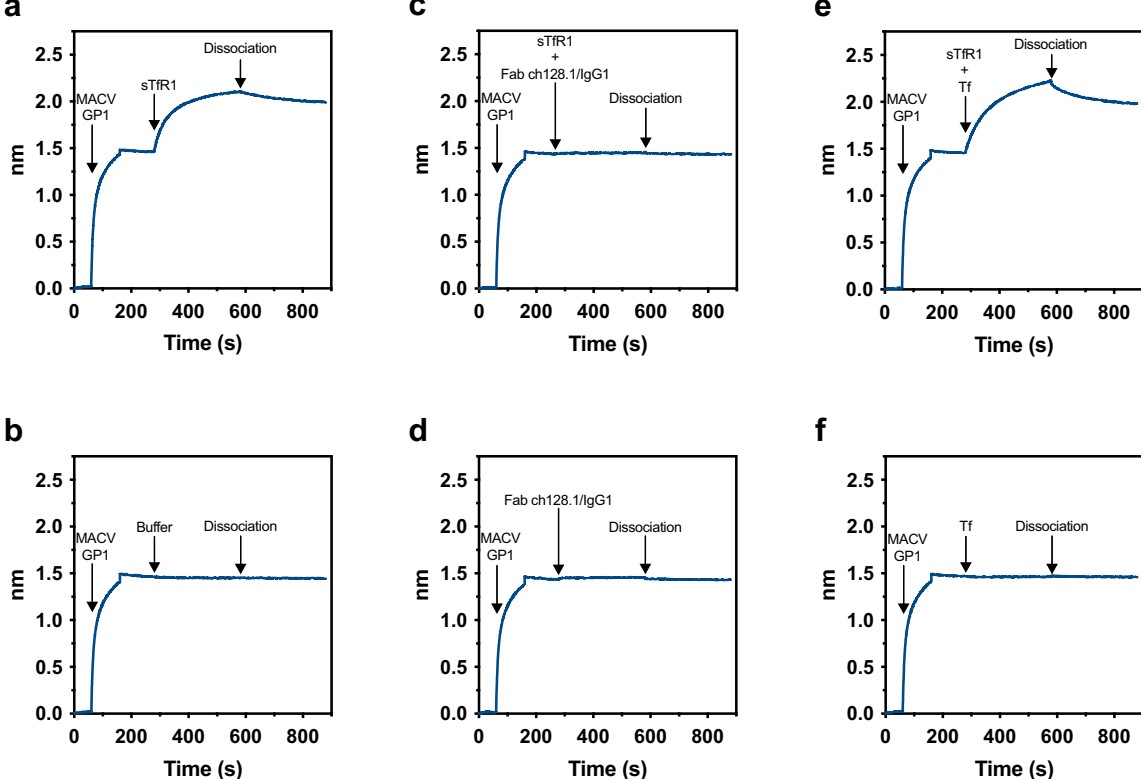

**Fig. 2 Assessment of Fab ch128.1/IgG1 and MACV GP1-Fc for binding to sTfR1 by bio-layer interferometry.** MACV GP1-Fc immobilized onto anti-human Fc biosensor tips and incubated with (**a**) sTfR1 or (**b**) buffer alone, (**c**) sTfR1 in complex with Fab ch128.1/IgG1 or (**d**) Fab ch128.1/IgG1 alone, or (**e**) sTfR1 in complex with Tf or (**f**) Tf alone. The arrows indicate the time points at which the indicated proteins were added and the dissociation step. The experiment was performed twice with representative data from one of the experiments shown. Seconds (s). Source data are included in the Source Data file.

against the Romero strain of JUNV was confirmed employing the same assay in which the $EC_{90}$ of the ch128.1/IgG1 and ch128.1/IgG1 mutant antibodies were 0.45 and 0.93 nM, with SI values of >2222 and >1008, respectively.

**Competitive inhibition of MACV GP1 binding by Fab ch128.1/IgG1.** We have previously shown in a cell-based assay that ch128.1/IgG3 competes with the MACV GP1 receptor-binding domain for hTfR1 expressed on HEK293T cells[10]. To further characterize the competitive interaction, we used bio-layer interferometry to assess the binding of a MACV GP1-Fc fusion protein to soluble human transferrin receptor 1 (sTfR1) in the presence of the Fab ch128.1/IgG1 or Tf (Fig. 2). While sTfR1 bound to immobilized MACV GP1-Fc (Fig. 2a), this interaction was not observed in the presence of the Fab ch128.1/IgG1 (Fig. 2c). In contrast, Tf, which does not compete with MACV GP1 for binding to sTfR1[8], did not affect sTfR1 binding to MACV GP1-Fc (Fig. 2e). The results show that Fab ch128.1/IgG1 blocks the binding of MACV GP1 to hTfR1 and suggest that MACV GP1 and ch128.1 antibodies bind to an overlapping region on hTfR1.

**Crystal structure of Fab ch128.1/IgG1 and computational docking to hTfR1.** The Fab fragment of the mouse/human chimeric 128.1 IgG1 antibody, Fab ch128.1/IgG1, was obtained through proteolytic digestion with papain and the Fc domain removed with immobilized protein A. Crystals of the Fab ch128.1/IgG1 grown by vapor diffusion diffracted to a resolution of 2.6 Å (Table 1). The structure contained Fab molecules in three

**Table 1 Data collection and refinement statistics for Fab ch128.1/IgG1.**

| Parameters | Value |
|---|---|
| Space group | $P3_1$ |
| Unit cell dimensions a, b, c | 126.27 Å, 126.27 Å, 94.85 Å |
| Unit cell dimensions $\alpha$, $\beta$, $\gamma$ | 90°, 90°, 120° |
| Wavelength | 0.978 Å |
| Resolution | 54.68 – 2.6 Å |
| I/σI | 11.34 |
| Completeness % | 99.63 % |
| Number of total reflections | 274,668 |
| Number of unique reflections | 51856 |
| CC1/2 | 99.6 |
| Reflections used in refinement | 51,844 |
| Reflections used for R-free | 5183 |
| R-work | 0.2072 (0.2715) |
| R-free | 0.2580 (0.3394) |
| Number of used reflections in refinement | 51,844 |
| Number of non-hydrogen atoms | 10,024 |
| Number of protein residues | 1290 |
| RMS (bonds) | 0.004 |
| RMS (angles) | 0.63 |
| *Ramachandran plot* | |
| Residues in favored region % | 95.62% |
| Residues in allowed region % | 4.15% |
| Residues in outlier region % | 0.23% |
| Disallowed % | 0% |

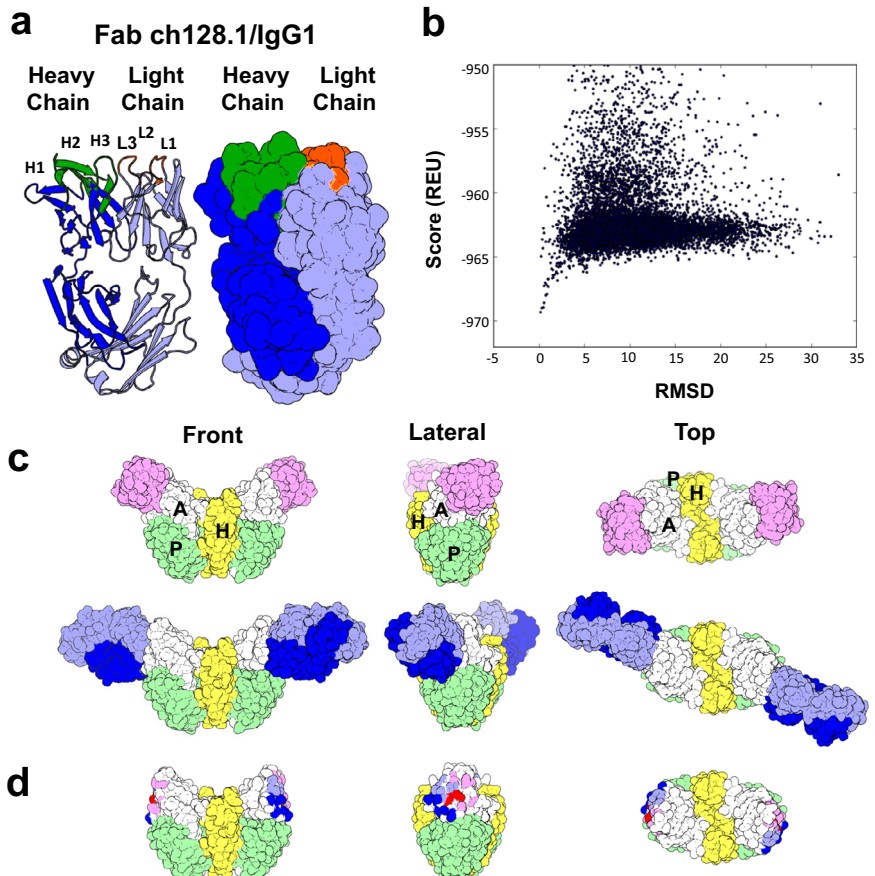

**Fig. 3 Crystal structure of Fab ch128.1/IgG1 and docking model of Fab binding to hTfR1. a** Structure of the Fab ch128.1/IgG1 highlighting the CDRs. On the left is a schematic representation of the antibody Fab fragment showing in light blue the light chain, in orange the position of the light chain CDRs (L1, L2, and L3), in blue the heavy chain and in green the heavy chain CDRs (H1, H2, and H3). On the right is a 3D surface representation of the Fab. The images were generated with PyMOL. **b** Computational modeling of Fab ch128.1/IgG1 binding to hTfR1. PyRosetta binding funnel generated from the largest cluster of docking models obtained with ClusPro. Each dot shows the PyRosetta binding score in Rosetta energy units (REU) and the RMSD from the initial configuration for the 17,003 docking models generated. **c** Surface representation of the MACV GP1 in complex with the hTfR1 crystal structure (PDB ID:3KAS) and of the Fab ch128.1/IgG1-hTfR1 docking model. Shown on top is the surface representation of MACV GP1 (pink) in complex with the hTfR1 with the helical (H, yellow), protease-like (P, green) and apical (A, white) domains in front view, rotated 90° on the y axis (lateral) and rotated 90° on the x axis (top). Below is the Fab ch128.1/IgG1-hTfR1 docking model with the same views with the heavy chain in blue and the light chain in light blue. It is important to note the steric overlap between the structures of the Fab ch128.1/IgG1 docking model and the MACV GP1 bound to the apical domain of hTfR1. Note that the Fab ch128.1/IgG1 CDRs are not represented by different colors in this panel. **d** Overlap of the MACV GP1 and the predicted Fab ch128.1/IgG1 footprints on hTfR1. Surface representation of hTfR1 with the front, lateral, and top orientations as in panel (**c**), showing on the apical domain the footprint of the MACV GP1 in pink and the footprint of the predicted Fab ch128.1/IgG1 docking model colored in blue for the heavy chain and light blue for the light chain. The overlapping contacts are shown in red.

distinct conformations within each asymmetric unit. All three conformations were nearly identical in their variable regions but differed at the linker with the constant region. The structure of Fab ch128.1/IgG1 shows amino acid residues 3–219 of the heavy chain that include the variable heavy chain and the $C_H1$ constant domain, and residues 3–211 of the light chain that include the variable light chain and the κ constant region (Fig. 3a). The structure included variable region frameworks and all six complementarity-determining regions (CDRs), including the residues in CDR-H1 between heavy chain (H) 26–35, CDR-H2 between H 50–66, CDR-H3 between H 99–107 and of residues in CDR-L1 between light chain (L) 23–33, CDR-L2 between L 49–55 and CDR-L3 between L 87–96 (Fig. 3a).

In previous studies, we have shown that ch128.1 binds to the extracellular domain of hTfR1 without interfering with the binding of its ligands Tf and HFE and that this interaction has an affinity to the apical domain of the receptor in the nanomolar range[10,21]. Importantly, this region overlaps with an area known to be critical for MACV GP1 binding and pathogenic NWM pseudovirus entry[9,10,16]. Using previous analysis as a basis for further modeling, we performed computational docking to generate a model of the Fab ch128.1/IgG1 bound to hTfR1. The initial orientation of the docking model was performed with the rigid body package, ClusPro[22]. The model was then fine-tuned through redocking with PyRosetta, a procedure that yielded a docking funnel with the best (i.e., most negative) scores, which fell into a narrow range of root mean square deviation (RMSD) relative to the starting conformation (Fig. 3b)[23,24]. This docking model was compared with the structure of a mammarenavirus GP1 bound to the apical domain of hTfR1[16]. Figure 3c shows a surface representation of the MACV GP1-hTfR1 cocrystal structure viewed from the front, lateral and top views, compared with the best docking model of Fab ch128.1/IgG1 to hTfR1. This in silico model predicts that Fab ch128.1/IgG1 and the MACV

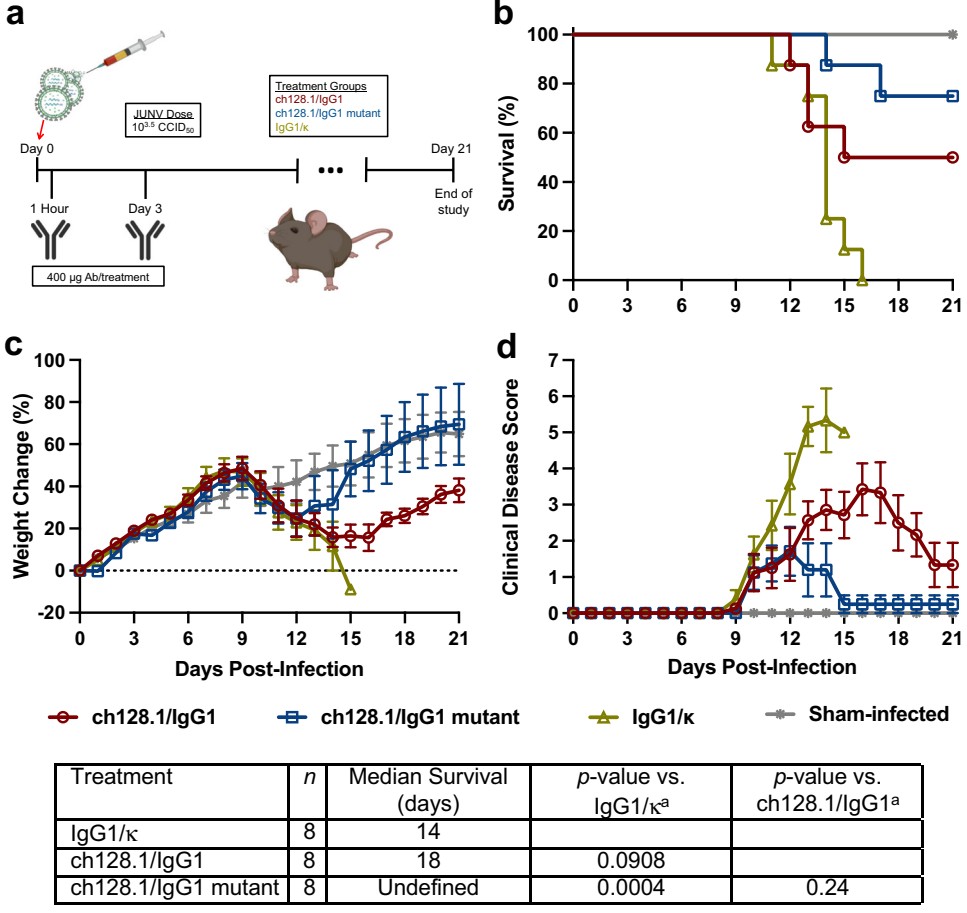

| Treatment | n | Median Survival (days) | p-value vs. IgG1/κ[a] | p-value vs. ch128.1/IgG1[a] |
|---|---|---|---|---|
| IgG1/κ | 8 | 14 | | |
| ch128.1/IgG1 | 8 | 18 | 0.0908 | |
| ch128.1/IgG1 mutant | 8 | Undefined | 0.0004 | 0.24 |

**Fig. 4 Study design and efficacy of anti-hTfR1 antibody treatment in JUNV-infected hTfR1 mice (Trial 1). a** Experimental design, (**b**) survival outcomes (n = 8 mice/treatment group and n = 3 untreated sham-infected controls), (**c**) percent change in body weight represented as a group mean ± standard error of the surviving animals compared to their starting weights on the day of infection and (**d**) clinical disease scores of surviving animals represented as group mean ± standard error during the course of the experiment. [a]Log-rank Mantel-Cox test. Antibody (Ab); Undefined: >50% survival. Source data are included in the Source Data file.

GP1 bind to an overlapping region on the hTfR1 apical domain (Fig. 3d). Based on this model and evidence for direct competition observed in bio-layer interferometry assays (Fig. 2), we conclude that Fab ch128.1/IgG1 sterically hinders binding of the MACV GP1 to the apical domain of hTfR1 and thereby inhibits TfR1-mediated NWM entry into human cells. We expect this competition to extend to other pathogenic NWMs given their binding to the same region of hTfR1[10,16].

**Evaluation of ch128.1/IgG1 and ch128.1/IgG1 mutant treatments in the hTfR1 mouse JUNV challenge model.** To evaluate the efficacy of the anti-hTfR1 antibodies in the hTfR1 mouse JUNV infection model, 3-week-old hTfR1 mice were challenged with a $10^{3.5}$ 50% cell culture infectious dose (CCID$_{50}$) of JUNV 1 h before receiving 400 μg of the specified antibody treatment (Fig. 4a). The animals received an additional treatment of 400 μg of antibodies on day 3 post-infection (p.i.). All eight mice that were administered the matched human isotype control IgG1/κ succumbed to the infection (Fig. 4b). However, a significant protective effect was observed in mice treated with ch128.1/IgG1 mutant, as 6/8 (75%) survived the JUNV challenge. Treatment with ch128.1/IgG1 resulted in 50% survival but did not differ significantly compared to the ch128.1/IgG1 mutant or isotype control treatments.

To assess morbidity, individual animal weights were measured during the course of the study and reported as percent change in body weight (Fig. 4c). By day 10 p.i., all the infected, antibody-treated groups were losing weight with the IgG1/κ group exhibiting the most dramatic weight loss through day 15 when the last remaining animal required euthanasia. The ch128.1/IgG1 mutant and ch128.1/IgG1 groups lost weight until day 12 and 14 p.i., respectively, followed by weight gain that was comparable with the sham-infected control group. The animals were also evaluated daily for clinical disease signs represented as total mean clinical disease scores (Fig. 4d). All infected groups developed observable disease signs by day 10 p.i. The IgG1/κ-treated mice presented with the highest clinical disease burden. The surviving ch128.1/IgG1 mutant antibody-treated animals were collectively in the best health at the end of the study period, while several of the ch128.1/IgG1-treated mice recovered more slowly but were gaining weight and were generally in good condition on day 21 p.i. (Fig. 4c, d).

A second experiment was conducted to confirm the improved survival outcome observed in the initial efficacy study, with several modifications to increase the rigor of the analysis. Most notably, the challenge dose was increased to $10^4$ CCID$_{50}$ of JUNV and a phosphate-buffered saline (PBS) vehicle placebo treatment was included to control for effects associated with the mice receiving antibody treatments (Fig. 5a). In addition, a group

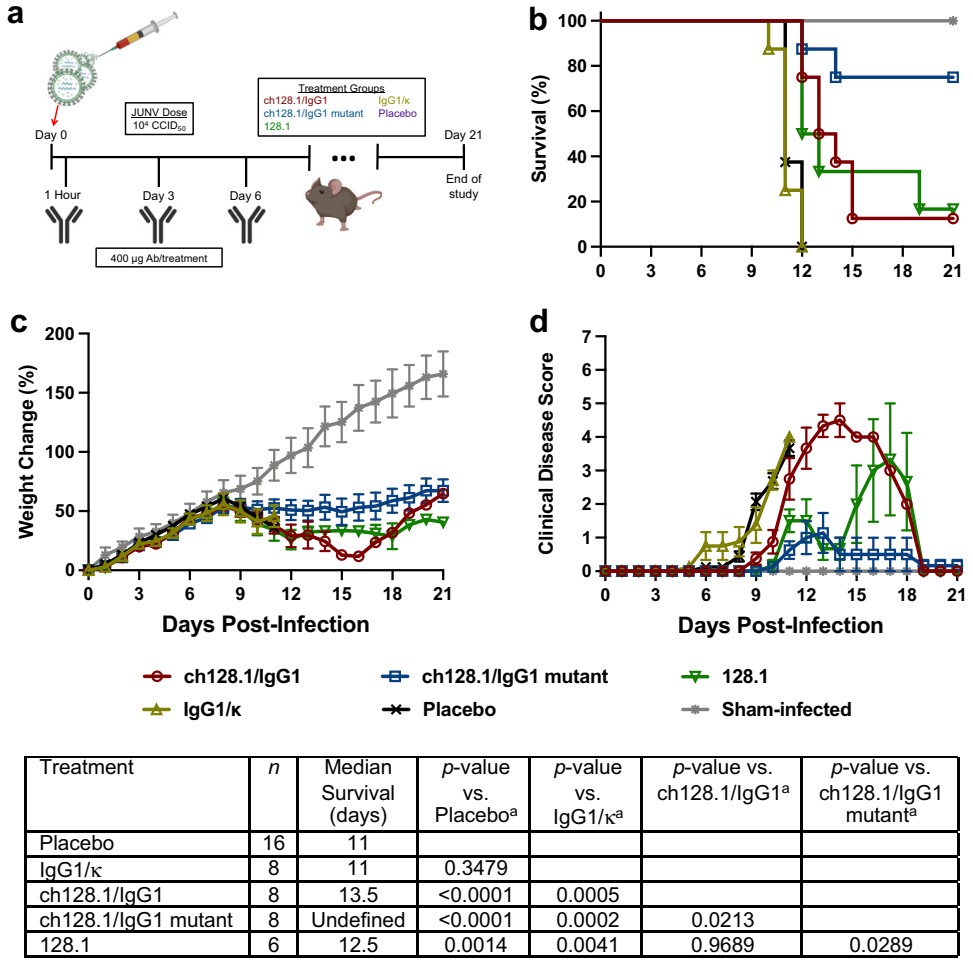

| Treatment | n | Median Survival (days) | p-value vs. Placebo[a] | p-value vs. IgG1/κ[a] | p-value vs. ch128.1/IgG1[a] | p-value vs. ch128.1/IgG1 mutant[a] |
|---|---|---|---|---|---|---|
| Placebo | 16 | 11 | | | | |
| IgG1/κ | 8 | 11 | 0.3479 | | | |
| ch128.1/IgG1 | 8 | 13.5 | <0.0001 | 0.0005 | | |
| ch128.1/IgG1 mutant | 8 | Undefined | <0.0001 | 0.0002 | 0.0213 | |
| 128.1 | 6 | 12.5 | 0.0014 | 0.0041 | 0.9689 | 0.0289 |

**Fig. 5 Study design and efficacy of anti-hTfR1 antibody treatment in JUNV-infected hTfR1 mice (Trial 2). a** Experimental design, **b** survival outcomes (n = 6–8 mice/treatment group, n = 16 placebo-treated mice, n = 3 untreated sham-infected controls), **c** percent change in body weight represented as a group mean ± standard error of the surviving animals compared to their starting weights on the day of infection, and **d** clinical disease scores represented as group mean ± standard error during the course of the experiment. [a]Log-rank Mantel-Cox test. Antibody (Ab); Undefined, >50% survival. Source data are included in the Source Data file.

receiving the 128.1 mouse IgG1 antibody was included for comparison and, to attempt to improve the efficacy of the anti-hTfR1 antibodies, a 3-dose treatment schedule was evaluated. Like the previous study, a significant protective effect was observed with the animals receiving ch128.1/IgG1 mutant antibody treatment with 75% (6/8) surviving the lethal JUNV challenge (Fig. 5b). Treatment with ch128.1/IgG1 or 128.1 showed similar results providing significant protection, although fewer survivors were observed and both treatment groups differed significantly compared to the ch128.1/IgG1 mutant treatment. By comparison, the IgG1/κ- and placebo-treated control mice all expired by day 12 p.i. (Fig. 5b). Mouse weights and clinical disease scores recorded during the study were consistent with the survival data with the animals treated with ch128.1/IgG1 mutant experiencing less severe clinical signs and better maintaining their body weights (Fig. 5c, d).

**ch128.1/IgG1 has minimal toxicity in hematopoietic progenitor cell cultures**. To evaluate the potential toxicity of ch128.1/IgG1 to normal cells expressing hTfR1, colony forming assays were conducted to determine the toxicity of the antibody to committed, human hematopoietic progenitor cells. These cells are known to express high levels of hTfR1, especially progenitors of the erythroid lineage in which high amounts of iron are

required for heme synthesis in the developing red blood cells[25–28]. This is in contrast to pluripotent, non-committed hematopoietic stem cells that show little to no expression of hTfR1[25,29,30]. A range of concentrations of ch128.1/IgG1 was tested and included the highest concentration of 75 μg/ml (500 nM). Figure 6 shows representative data from a single donor of bone marrow mononuclear cells (BMMC). There was a dose-response effect and a decrease in colony numbers was generally observed. However, this difference was only significant for the BFU-E (burst forming unit-erythroid) cells treated with 75 μg/ml ch128.1/IgG1. Similar results were observed with BMMC from two additional donors (Supplementary Fig. 1). Treatment with 500 nM staurosporine as a toxic compound control resulted in a complete lack of colony growth. These results suggest that there may be some potential toxicity to committed hematopoietic progenitor cells with high concentrations of ch128.1/IgG1, although colonies of all types, even those in the erythroid lineage, were still able to grow in all the tested conditions.

**ch128.1/IgG1 only partially impacts ferritin binding and internalization**. To investigate the possible inhibition of hTfR1 binding and the subsequent internalization of two hTfR1 ligands, Tf and heavy chain ferritin (H-Ft), by ch128.1/IgG1, we conducted competitive binding and internalization studies by

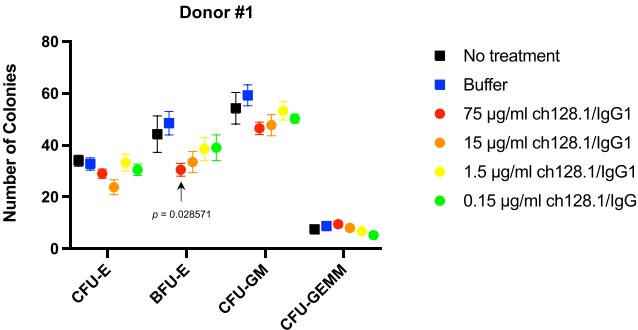

**Fig. 6 Evaluation of the potential toxicity of ch128.1/IgG1 to committed hematopoietic progenitor cells.** Human BMMC were treated with ch128.1/IgG1 at various concentrations and cultured for 14 days. Colony number data represent the mean ± standard error of quadruplicate samples. The data are representative of three separate experiments using BMMC from three different donors (data from donors two and three are shown in Supplementary Fig. 1). Statistical significance based on comparison to buffer alone treatment was determined using the nonparametric, unpaired Mann–Whitney test and the False Discovery Rate (FDR) approach with two-stage linear step-up method of Benjamini, Krieger, and Yekutieli was applied to account for multiple comparisons. Colony forming unit-erythroid (CFU-E), burst forming unit-erythroid (BFU-E), colony forming unit granulocyte/macrophage (CFU-GM), colony forming unit granulocyte/erythroid/macrophage/megakaryocyte (CFU-GEMM). Source data are included in the Source Data file.

standard flow cytometry and imaging flow cytometry. For binding experiments, we used erythroblasts that are known to express very high levels of hTfR1[28]. Figure 7a shows that ch128.1/IgG1 does not inhibit the binding of Tf to hTfR1, as demonstrated by the similarity of the fluorescence signal in cells incubated with and without ch128.1/IgG1. This is consistent with previous reports that show antibodies sharing the same variable region as ch128.1/IgG1 do not inhibit Tf binding to hTfR1[26,31].

The competitive binding assay using H-Ft showed that ch128.1 partially inhibits the binding of H-Ft to hTfR1 on the surface of erythroblasts, as evidenced by a decrease in fluorescent signal in the presence of ch128.1/IgG1 (Fig. 7b, c). However, this inhibition was far less than that observed with the murine anti-hTfR1 antibody clone M-A712, another hTfR1 binding antibody targeting the apical domain that is known to block H-Ft binding to hTfR1[32,33] (Fig. 7c). The M-A712 antibody shows a more substantial decrease in fluorescent signal compared to ch128.1/IgG1. Figure 7 panels 7a and 7b show data using erythroblasts that had been in culture for 13 days. Figure 7c represents data from a replicate H-Ft binding experiment using erythroblasts cultured for 22 days and includes the M-A712 antibody. The variability of staining between Fig. 7b and 7c are due to the assays being conducted at different times during the erythroblast culture. These cultures contain erythroid progenitors at various stages of differentiation and thus contain cells expressing various levels of hTfR1. It is important to note that both Fig. 7b and 7c show that ch128.1/IgG1 only partially inhibits H-Ft binding.

To have a more homogenous population of cells, the internalization of Tf was evaluated using human MM.1S cells that also express high levels of hTfR1. As measured by standard flow cytometry, Fig. 8a shows increasing internalization of Tf by MM.1S cells over time, which is unaffected by the presence ch128.1/IgG1. Figure 8b displays these data in the form of a line graph of median fluorescence intensities. To further assess internalized Tf from that bound to hTfR1 on the cell surface and the effect of ch128.1/IgG1 on the internalization process, the same samples were analyzed by imaging flow cytometry.

The resulting images were analyzed to exclude fluorescence localized to the cell membrane. Figure 8c shows representative brightfield and fluorescence images of MM.1S cells incubated at 37 °C for 60 min and confirms Tf internalization in the presence of ch128.1/IgG1. Figure 8d is the histogram of fluorescence intensity (excluding the cell membrane fluorescence) of samples incubated at 37 °C for 60 min. Figure 8e is a line graph that plots the geometric mean fluorescence intensities (excluding cell membrane fluorescence) of MM.1S cells incubated with the IgG1/κ isotype control or ch128.1/IgG1 over time. Collectively, these results confirm that the internalization of Tf is not affected by ch128.1/IgG1.

The internalization of H-Ft was also evaluated using standard flow cytometry. Figure 9a shows that the internalization of H-Ft by MM.1S cells increases over time and that this internalization is partially inhibited in the presence of ch128.1/IgG1. Figure 9b displays these data in the form of a line graph of median fluorescence intensities. The inhibition of H-Ft uptake by ch128.1/IgG1 is considerably less than the inhibition seen in the presence of the murine anti-hTfR1 antibody M-A712. Furthermore, we used imaging flow cytometry to focus the analysis on fluorescence of internalized H-Ft (as done for Tf internalization). Figure 9c contains representative brightfield and fluorescence images of MM.1S cells incubated at 37 °C for 60 min and confirms that H-Ft is internalized. Figure 9d shows the histogram of the imaging flow cytometry analysis in cells incubated at 37 °C for 60 min. Figure 9e plots the geometric mean fluorescence intensities (excluding fluorescence of the cell membrane). These results are further evidence that internalization of H-Ft is only partially inhibited by ch128.1/IgG1 compared to the IgG1/κ isotype control; however, this inhibition is not as dramatic as that observed with the M-A712 antibody.

## Discussion

The pathogenic NWMs bind to a common region on the apical domain of hTfR1[9], which does not overlap with the docking region of the two major physiological ligands of hTfR1, Tf and HFE[13,15,34]. Thus, targeting the apical domain of hTfR1 with an antibody may be an effective therapeutic strategy to broadly inhibit infection without substantially interfering with cellular iron metabolism. Towards this goal, ch128.1/IgG3 had previously been shown to inhibit entry by pseudotyped viruses expressing the envelope glycoproteins of pathogenic NWMs and the attenuated JUNV IV445 strain in cell culture[10]. In contrast to human IgG3, human IgG1 has longer bioavailability in blood, is less prone to proteolysis, is less immunogenic and has an established manufacturability[35–38]. Therefore, we evaluated and confirmed the antiviral activity of ch128.1/IgG1 against vaccine and pathogenic strains of JUNV as a prerequisite to in vivo efficacy studies. However, the evaluation of this novel host receptor-targeted therapeutic strategy in vivo presented significant challenges as the only immune-competent laboratory rodent species known to be susceptible to JUNV, MACV, and Guanarito virus (GTOV) infections is the guinea pig[39–41], and based on the amino acid sequence of the apical domain of the guinea pig TfR1, the ch128.1 antibody would not be expected to cross-react. By contrast, cross-reactivity has been demonstrated with TfR1 orthologs from the Old World rhesus macaques (*Macaca mulatta*) and African green monkeys (*Cercopithecus aethiops*)[10], but due to high cost and relative inaccessibility of primate models for early stages of antiviral drug development, a small-animal disease model is better suited for proof-of-concept studies.

We recently developed a hTfR1 mouse model of JUNV infection and disease[20]. In addition to its value for the study of JUNV pathogenesis, this is the only characterized small-animal model of

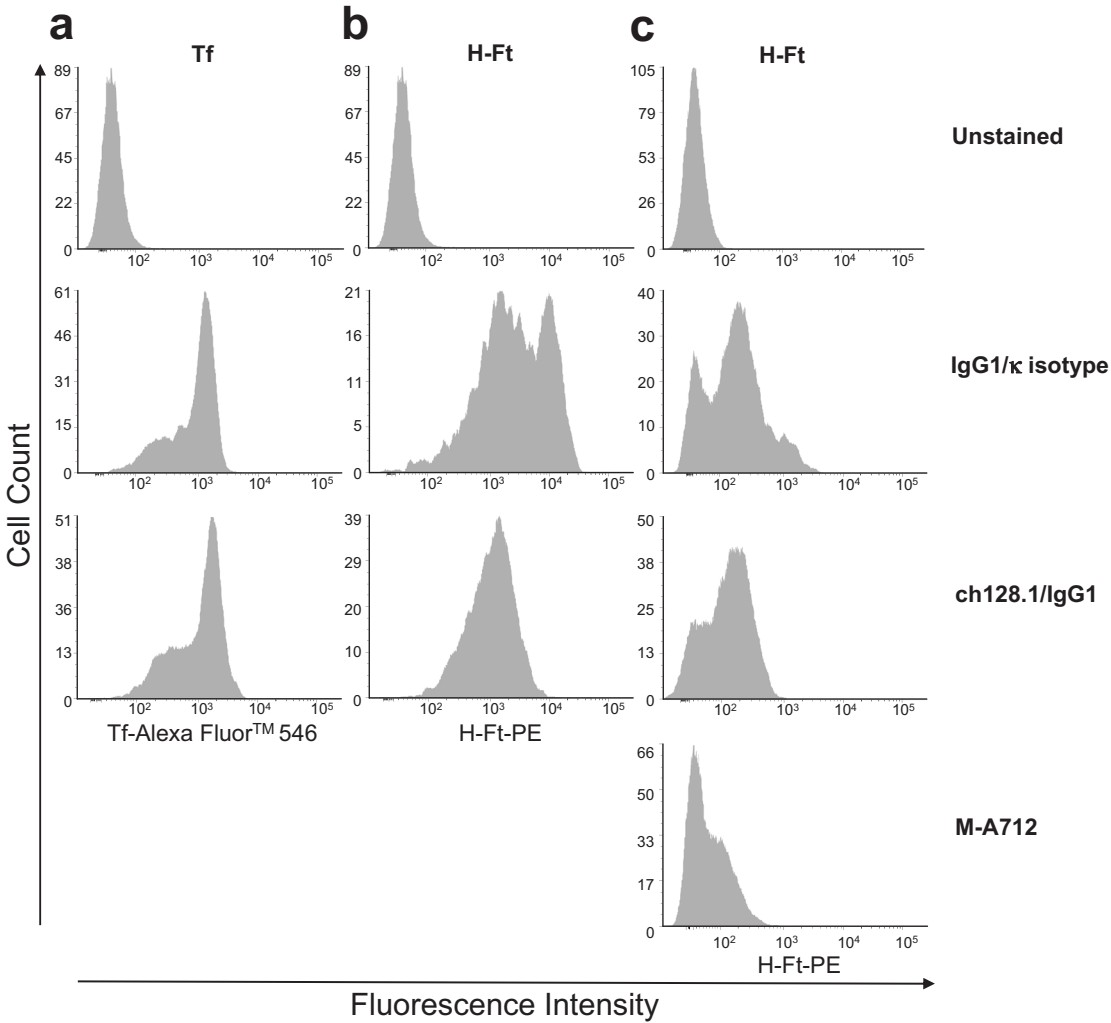

**Fig. 7 Flow cytometric analysis of Tf and H-Ft binding to hTfR1 in the presence of ch128.1/IgG1.** Erythroblasts, which are known to express high levels of hTfR1, were differentiated and cultured from bone marrow CD34+ cells. The erythroblasts were incubated on ice with (**a**) a combination of Tf and an IgG1/κ isotype control or ch128.1/IgG1, (**b**) a combination of H-Ft and IgG1/κ isotype control or ch128.1/IgG1 or (**c**) a repeat of the H-Ft binding experiment that included the anti-hTfR1 antibody M-A712. Histograms represent the fluorescence intensity due to the binding of Tf or H-Ft to hTfR1 on the surface of the cells.

NWM infection in which hTfR1-directed therapeutics can be evaluated. Further, the mouse model is ideal for proof-of-concept studies as larger numbers of animals can be used, less drug is required per animal and a wealth of mouse reagents are available to support additional downstream analyses. Using the new hTfR1 mouse model of NWM infection and disease, we demonstrated for the first time that a therapeutic antibody directed at the apical domain of hTfR1 can protect against lethal JUNV disease in vivo. Efficacy was assessed for both ch128.1/IgG1 and its Fc mutant counterpart in two separate studies. In both experiments, the ch128.1/IgG1 mutant antibody, which has impaired binding to Fc gamma receptors (FcγRs) and C1q resulting in lack of effector functions (antibody-dependent cell-mediated cytotoxicity (ADCC), antibody-dependent cell-mediated phagocytosis (ADCP), and complement-dependent cytotoxicity (CDC)), proved to be the most efficacious in limiting severe JUNV disease with a high level of protection achieved. It is possible that due to the ubiquitous expression of hTfR1[11,12,42], complete protection would require higher doses and/or more frequent administration of the anti-hTfR1 antibodies. Future studies could explore other treatment regimens aimed at improving efficacy. Additional studies would

also be required to assess whether relapse during convalescence to the late neurologic syndrome (LNS) observed in approximately 10% of individuals treated with immune plasma[43] can be reproduced in the hTfR1 mouse model and, if so, if treatment with the anti-hTfR1 antibodies would reduce the risk of relapse.

Administration of the ch128.1/IgG1 mutant was more effective than its wild-type counterpart ch128.1/IgG1 in combating infection in our initial comparative studies. These results are consistent with the idea that, with the tested treatment regimens, the Fc effector functions are not necessary for protection based on competitive inhibition of NWM binding to hTfR1. The greater protective effect conferred by the ch128.1/IgG1 mutant might be attributed to potential Fc-mediated adverse effects of the wild-type IgG1 version of the antibody[44]. However, in preliminary toxicity studies using heterozygous $hTfR1^{+/-}$ mice, we found that both antibodies were equally well-tolerated. It is also conceivable that free antibodies more effectively compete with JUNV compared to those bound to FcγRs present on effector cell populations. Interestingly, the original mouse 128.1 IgG1 antibody exhibited a level of protection similar to the ch128.1/IgG1 antibody; but in contrast to human IgG1, murine IgG1 does not

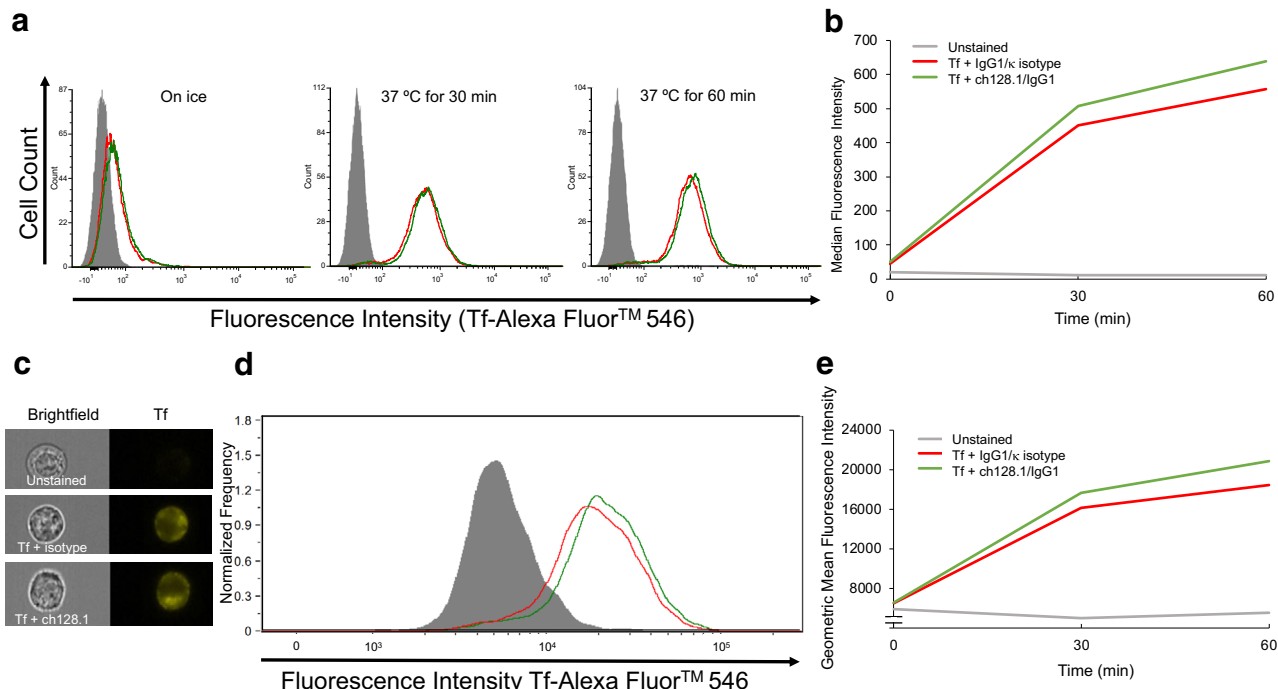

**Fig. 8 Internalization of Tf into MM.1S cells in the presence of ch128.1/IgG1. a**, **b** Standard and (**c**–**e**) imaging flow cytometric analysis of Tf internalization in the presence of the IgG1/κ isotype control (red line) or ch128.1/IgG1 (green line). Negative control unstained cells (gray-filled) are also shown. **a** MM.1S cells incubated on ice for 60 min, after 30 min at 37 °C and after 60 min at 37 °C. **b** Median fluorescence intensities of Tf fluorescence over time. **c** Representative images of cells incubated for 60 min at 37 °C. **d** Internal fluorescence intensities of imaging flow cytometry at 37 °C for 60 min. **e** Geometric mean fluorescence intensities of internalized Tf over time.

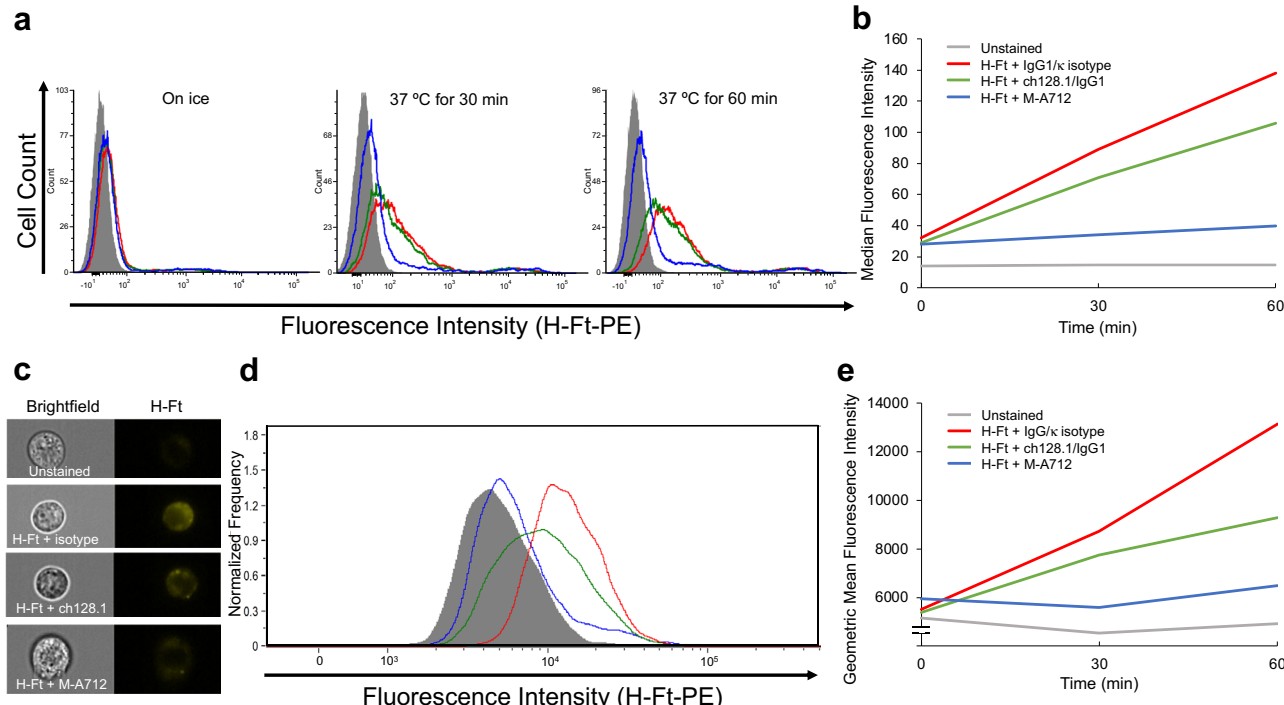

**Fig. 9 Internalization of H-Ft into MM.1S cells in the presence of ch128.1/IgG1. a**, **b** Standard and (**c**–**e**) imaging flow cytometric analysis of H-Ft internalization in the presence of the IgG1/κ isotype control (red line), ch128.1/IgG1 (green line) or the anti-hTfR1 antibody M-A712 (blue line). Negative control unstained cells (gray-filled) are also shown. **a** MM.1S cells incubated on ice for 60 min, after 30 min at 37 °C and after 60 min at 37 °C. **b** Median fluorescence intensities of H-Ft fluorescence over time. **c** Representative images of cells incubated for 60 min at 37 °C. **d** Internal fluorescence intensities of imaging flow cytometry analysis at 37 °C for 60 min. **e** Geometric mean fluorescence intensities of internalized H-Ft over time.

activate complement and does not bind the high affinity FcγRI, although it binds FcγRIIb and FcγRIII[45–48]. Further studies are needed to better define the efficacy of these anti-hTfR1 antibodies and understand the differences in the level of protection.

Knowledge of the molecular basis for the interaction between the variable region of the ch128.1 antibody and its epitope provides a foundation for improving the targeting and optimization of virus blockade. The crystallographic structure of Fab ch128.1/IgG1 also allows for the rational optimization of its framework and CDRs. The result of our docking analysis of the Fab ch128.1/IgG1 with hTfR1 predicts the footprint on the apical domain of the receptor that overlaps with the residues responsible for the interaction of MACV GP1 with hTfR1[16]. This steric hindrance offers a general mechanism underlying the blockage of entry by the pathogenic NWMs and facilitates the rational design of therapeutic agents with similar modes of action. Importantly, this model of inhibition is supported by previous studies evaluating binding of ch128.1/IgG3 and a MACV GP1 fusion protein to human cells by flow cytometry[10] and the present protein biochemical assays using bio-layer interferometry, both demonstrating competition between ch128.1 and MACV GP1 for the apical domain of hTfR1.

With any therapeutic approach that targets the host, unintended consequences must be considered. Potential toxicity associated with treatment was assessed in primary human cells that express high levels of hTfR1. Only erythroid progenitors were affected by ch128.1/IgG1 treatment as evidenced by the slight but significant decrease of BFU-E colonies. This is logical due to the fact that these normal erythroid progenitor cells express the highest level of hTfR1 in the body[28]. However, it is important to note that ch128.1/IgG1 treatment did not completely inhibit erythroid colony formation, even at the highest concentration tested, suggesting that this population of cells would not be adversely affected in patients undergoing therapy. Moreover, pluripotent non-committed hematopoietic stem cells show little to no expression of hTfR1[25,29,30]. Even treatment with a fusion protein consisting of an IgG3 version of ch128.1 and avidin conjugated to a biotinylated version of the plant toxin saporin is not toxic to these pluripotent stem cells[49]. Thus, any erythroid progenitors that are lost can be repopulated by the stem cell compartment. It is also important to note that this in vitro assay may not reflect in vivo conditions. In this assay, a very high concentration of the antibody was incubated with the cells for 14 days. In an animal or a patient, the half-life and bioavailability of the antibody would also be expected to affect toxicity; thus the limited toxicity observed in our studies may overestimate the impact of treatment on these cell populations in patients.

Other anti-hTfR1 antibodies have been reported to inhibit colony formation in early studies where erythroid progenitors were not determined and only the colony forming unit granulocyte/macrophage (CFU-GM) colony type was reported. These included the mouse anti-hTfR1 IgG1 antibodies B3/25 (non-neutralizing, meaning it does not inhibit the binding of Tf to TfR1) and 43/31 (neutralizing, blocks the binding of Tf), as well as the mouse anti-hTfR1 IgA antibody 42/6 (inhibits Tf binding through steric hindrance)[26,50]. In a later study where erythroid colonies were enumerated, the mouse IgG1 antibody A24 (neutralizing antibody) was reported to decrease colony forming unit-erythroid (CFU-E) colony numbers at a concentration of 10 μg/ml[51]. This study also showed an increase in CFU-GM, suggesting that the A24 antibody induced a commitment toward the monocyte lineage versus the granulocyte lineage. A non-neutralizing anti-hTfR1 IgM antibody (RI7-208) was also reported to decrease CFU-E numbers in the bone marrow of mice given 1 mg of the antibody daily for 7 days[52]. Furthermore, the fully human neutralizing anti-hTfR1 antibody JST-TFR09 (also

known as PPMX-003) demonstrated toxicity to erythroblasts differentiated in vitro from CD34$^+$ cells using concentrations above 156 ng/ml[53]. Moderate anemia has been observed in cynomolgus macaques given multiple doses of the JST-TFR09 antibody[53,54]. Taken together, these studies suggest that the erythroid progenitors are vulnerable to the toxic effects of anti-TfR1 antibodies and that the degree of vulnerability depends partly on whether the antibody is a neutralizing antibody or not. In general, neutralizing antibodies show more toxicity than their non-neutralizing counterparts. ch128.1/IgG1 is a non-neutralizing antibody and its toxic effect on the erythroid progenitor population seems to be minimal. Further studies in relevant animal models and humans are needed to determine the effect of ch128.1/IgG1 on normal cells.

It is a logical concern that ch128.1/IgG1 will compete with the binding of known TfR1 ligands, Tf, HFE and H-Ft, and therefore affect downstream processes such as receptor internalization and iron uptake. The variable region of ch128.1 binds with high affinity (5.7 nM) to the hTfR1 between residues S324 to S368 within the apical domain of the receptor[10]. Here, we show that ch128.1 does not inhibit the binding or internalization of Tf, which binds at the basal portion of hTfR1 formed by the helical and protease-like domains close to the cell membrane[14,15,55]. This was expected since we have previously shown that antibodies with the same variable region as ch128.1/IgG1 do not inhibit Tf binding[26,31]. HFE binds hTfR1 in the helical domain in the dimer interface region, which overlaps with the binding site of Tf[13–15,56]. HFE does not impair the binding of H-Ft[33]. Additionally, we have previously shown that HFE does not compete with antibodies containing the ch128.1 variable region for binding to hTfR1[21]. Therefore, ch128.1/IgG1 should not interfere with the binding of HFE to TfR1. However, ch128.1/IgG1 partially inhibits the binding and internalization of H-Ft. The TfR1 binding site for ch128.1/IgG1 partially overlaps with certain residues where H-Ft binds, specifically hTfR1 residues E343, K344, and N348[14]. This could explain the partial inhibition of H-Ft internalization by ch128.1/IgG1. The murine anti-hTfR1 M-A712 antibody also shares binding sites with H-Ft, with M-A712 binding to a region on hTfR1 that includes residues R208-L212[32], an area shown to include binding sites for H-Ft[14]. Since Tf is the main iron transporter and ch128.1/IgG1 does not inhibit its binding or internalization, combined with the fact that H-Ft has been shown to only bind cells expressing a very high density of hTfR1[14], we expect the impact of ch128.1/IgG1 on iron uptake in most normal cells to be minimal.

The severity of disease caused by NWM infection and the limited countermeasures available underscore the urgent need for effective and broadly active therapeutic interventions. The host receptor-targeted therapeutic strategy of interfering with the ability of the pathogenic NWMs to bind their host cell receptor generally avoids concerns with the emergence of drug resistance that is common with direct-acting antivirals that target the virus. Recently, the development of an immunoadhesin (Arenacept) with the sequence of white-throated woodrat TfR1 apical domain fused to the Fc region of human IgG1 was reported[57]. Arenacept inhibits cellular entry of pseudotyped viruses expressing MACV, JUNV, GTOV, or Sabiá virus envelope glycoproteins, as well as native JUNV and MACV infection by standard plaque reduction assays[57]. An aptamer targeting the apical domain of hTfR1 has also been described and found to be active in cell culture-based assays[58]. More recently, the murine antibody OKT9, also targeting the apical domain of hTfR1, was shown to inhibit pathogenic NWM entry in vitro[59]. The ch128.1/IgG1 mouse/human chimeric antibody, the OKT9 mouse antibody, the aptamer, and the Arenacept strategies all exploit the vulnerability of NWM reliance on hTfR1. However, only the ch128.1/IgG1 antibody

approach reported here has been proven in vivo to protect against lethal disease caused by a pathogenic NWM and is ready for human use. Our encouraging results warrant further studies in support of therapeutic development of the ch128.1/IgG1 antibody or related derivatives aimed at disrupting the interaction of the pathogenic NWM GP1 subunit with their cognate host receptor. Ultimately, the use of ch128.1/IgG1 with other promising direct-acting small molecule antivirals[60] or antibodies targeting the viral envelope glycoprotein[61,62] would provide a complementary therapeutic strategy that would increase efficacy and reduce the emergence of drug resistance.

## Methods

**Cells and animals for virus infection studies.** A549 human epithelial lung (ATCC® CCL-185™) and Vero African green monkey kidney (ATCC® CCL-81™) cells were purchased from American Tissue Culture Collection (Manassas, VA, USA) and maintained in Dulbecco's modified eagle medium (DMEM) and minimal essential medium (MEM), respectively. The medium was supplemented with 5% fetal bovine serum (FBS; HyClone, UT, USA).

C57BL/6 hTfR1 knock-in (human *TFRC* replacing the mouse *TFRC*) mice were obtained from Genentech, Inc. (San Francisco, CA, USA) and have been previously described[42]. Heterozygous hTfR1$^{+/-}$ mice were bred to produce homozygous hTfR1$^{+/+}$ founders. The founding animals were backcrossed twice with AG129 mice deficient in type I and type II IFN receptors (IFN-α/βγR$^{-/-}$). The development of mice on a hybrid background was a consequence of the genetic characterization of hTfR1 mice with various type I and/or type II interferon receptor deficiencies[20]. For the studies described here, mice homozygous for hTfR1 and intact type I and type II IFN receptors (IFN-α/βγR$^{+/+}$) were produced and confirmed by PCR genotyping for human and mouse TfR1 and the presence of IFN-α/β and IFN-γ receptors. Mice were housed in a GM500 Green Line IVC system (Tecniplast SpA, Italy) in individually ventilated cages and fed Harlan Lab Block and tap water ad libitum. Room air temperature in the biosafety level-3 enhanced laboratory dedicated to JUNV work was 72 ± 4 °F with 30–70% air humidity. The room had a 12–12 h dark/light cycle.

**Viruses.** The Candid#1 JUNV live-attenuated vaccine strain (JUNV-C1; 1 passage in BS-C-1, two passages in Vero cells) was kindly provided by Dr. Robert Tesh (World Reference Center for Emerging Viruses and Arboviruses, University of Texas Medical Branch (UTMB), Galveston, TX, USA). The molecular clone of the pathogenic Romero strain of JUNV was generously provided by Dr. Slobodan Paessler (UTMB). The recombinant JUNV was rescued in BHK-21 cells as previously described[63], and the stock was prepared from a single passage in Vero cells and sequenced to verify accuracy. For JUNV challenge studies, the virus stock was diluted in sterile MEM to achieve the desired concentrations for infection. Sham infections consisted of MEM alone. All work with the pathogenic JUNV strain Romero was conducted in enhanced biosafety level-3+ containment by Candid#1-vaccinated personnel at Utah State University (Logan, UT, USA).

**Antibodies.** The antibody 128.1 is a murine monoclonal IgG1 specific for hTfR1[64]. The mouse/human chimeric IgG1/κ antibody (ch128.1/IgG1) contains the variable regions of the murine antibody 128.1[65]. The mutant version of this chimeric antibody (ch128.1/IgG1 mutant) contains the mutations L234A, L235A, and P329S in the γ1 heavy chain[65]. This triple mutant antibody was designed to impair binding to FcγRs and to the complement component C1q, which results in impaired ADCC, ACDP, and CDC[35,44,65,66]. A mouse/human chimeric IgG1/κ antibody specific for the hapten dansyl (5-dimethylamino naphthalene-1-sulfonyl chloride)[67] was used as an isotype control antibody (IgG1/κ). All three antibodies were produced in murine myeloma cells. The hybridoma expressing the 128.1 antibody and the transfectomas expressing the chimeric antibodies were grown in roller bottles and antibodies were purified using affinity chromatography as described[21,65–68].

**Inhibition of JUNV infection by ch128.1/IgG1 and ch128.1/IgG1 mutant in cell culture.** The antiviral activity of ch128.1/IgG1 and ch128.1/IgG1 mutant antibodies against JUNV Romero and JUNV-C1 was evaluated in A549 cells by virus yield reduction assay[69]. Briefly, subconfluent cells in 96-well microplates were treated for 1 h with half-log$_{10}$ dilutions of antibodies in culture media supplemented with 2% FBS before being overlayed with an inoculum containing JUNV Romero or JUNV-C1 at a multiplicity of infection (MOI) of 0.001. Plates were incubated for 4 days at 37 °C and 5% CO$_2$ before culture supernatants were titrated to detect the presence and concentration of JUNV by endpoint dilution on Vero cells as previously described[69]. Briefly, a specific volume of culture supernatant was serially diluted and added to triplicate wells of Vero cell monolayers in 96-well microtiter plates. The viral cytopathic effect was determined 11 days after plating and the 50% endpoints were calculated by the method of Reed and Muench[70]. The lower limit of detection (LLD) for culture supernatant was 1.67 log$_{10}$ CCID$_{50}$/ml. The 50% cell cytotoxic dose (CC$_{50}$)

was determined by neutral red dye uptake in uninfected cells treated with the antibodies and cultured in parallel and was >1000 nM for both antibodies. The 90% effective concentration (EC$_{90}$) was determined by regression analysis and represents the concentration of antibody that reduced the virus yield by one log$_{10}$. The SI was calculated using the formula: SI = CC$_{50}$/EC$_{90}$.

**Preparation of Fab ch128.1/IgG1 fragments.** The ch128.1/IgG1 antibody (2 mg) was digested at 37 °C with 0.05 mg of the endopeptidase papain immobilized on agarose, and the Fc domain was removed with immobilized Protein A using the Pierce™ Fab Preparation Kit (Thermo Fisher Scientific, Waltham, MA, USA, Cat. No. PI44985) as specified by the manufacturer. For crystallization studies, the eluted Fab fraction was further purified through size exclusion chromatography (SEC) using a high-resolution ENrich SEC 650 10 × 300 mm column (Bio-Rad Laboratories, Hercules, CA, USA) with a medium pressure NGC Quest™ 100 Chromatography System (Bio-Rad Laboratories). Prior to purification, the column was washed with two volumes of deionized water and two volumes of running buffer (50 mM Tris-Cl, 150 mM NaCl, pH 7.6) using a flow of 1 ml/min. The 2 ml fraction with the Fab ch128.1/IgG1 was concentrated to 500 µl using an Amicon® Ultra-4 10 K Centrifugal Filter Device (MilliporeSigma, St. Louis, MO, USA) with 10 kDa pore size and large aggregates and debris removed with a 0.22 µm Corning® Costar® Spin-X® centrifuge tube filter (MilliporeSigma). The concentrated sample was injected into the chromatography system and subsequently eluted using 1.5 volumes of running buffer. Fractions of 1.5 ml were collected and the purity, correct assembly and molecular weight were confirmed by SDS-PAGE.

**Production of sTfR1 and MACV GP1-Fc fusion protein.** DNA encoding human sTfR1 (GenBank: NM_003234.3 residues 121–760) was cloned into the pHLsec expression vector[71] downstream of the pHLsec secretion signal. sTfR1 was produced using linear polyethylenimine (PEI) to transfect HEK 293 S GnTI$^{-/-}$ cells (ATCC® CRL-3022™) grown in suspension culture and maintained in Free-Style 293 Expression Medium (Cat. No. 12338018) supplemented with 2% Ultralow IgG FBS (Cat. No. 16250078) and penicillin/streptomycin (Thermo Fisher Scientific). The supernatant was harvested 72 h post-transfection and sTfR1 purified using human Tf affinity chromatography as previously described[16,34]. sTfR1 eluted as a single peak in buffer containing 25 mM Tris-HCl (pH 7.5) and 150 mM NaCl by SEC on a Superdex® 200 Increase column (GE Healthcare Life Sciences, Marlborough, MA, USA).

The DNA encoding the MACV GP1 subunit (GenBank: NC_005078 residues 87–250) was cloned with a C-terminal Fc tag into pVRC8400 vector containing the Fc region of human IgG1[16]. The resulting MACV GP1-Fc fusion protein was expressed in HEK 293 T cells grown in suspension and purified by Protein A affinity chromatography according to the manufacturer's protocol (Thermo Fisher Scientific), followed by size exclusion chromatography on a Superdex® 200 Increase column, with the protein eluting at the expected retention time.

**Bio-layer interferometry competition assay.** All assays were performed using an Octet RED96 system (ForteBio, Fremont, CA, USA). For MACV GP1-Fc competition experiments, MACV GP1-Fc was loaded onto anti-human IgG Fc capture biosensors (ForteBio) at 17 nM for 100 s in kinetic buffer PBS containing 0.02% (v/v) Tween and 0.1% (w/v) bovine serum albumin. After equilibration in the buffer for 120 s, sTfR1 alone at 1.5 µM or in the presence of soluble ligands (ch128.1 Fab or Tf) at 2 µM were associated for 300 s. This was followed by a 300 s dissociation step.

**Crystallization and data collection of Fab ch128.1/IgG1.** The Fab ch128.1/IgG1 was crystallized by vapor diffusion in hanging drops at 18 °C with 576 different crystallization conditions. Wizard™ (Rigaku, WA, USA) and AmSO$_4$ (Qiagen, Hilden, Germany) 96-well plates were used with three conditions per well, obtaining crystals in several of the tested solutions. The crystals usually appeared within 1 to 5 weeks. Single crystals were frozen in liquid nitrogen, and their X-ray diffraction patterns were obtained using the 24-ID-E beamline with an energy range of 12.68-keV of the APS synchrotron (Advanced Photon Source, Argonne National Laboratory, Lemont, IL, USA). The structure of Fab ch128.1/IgG1 (Worldwide Protein Data Bank accession code PDB ID 6WLA[72]) was determined by molecular replacement using the Phaser (version 2.7.16) suite of programs. Crystallographic refinement was performed using Phenix version 1.8.4[73] and Buster version 2.11.2[74], while structure visualization and modeling were performed in COOT version 0.9.1[75] and PyMOL version 2.3.4 (Schrödinger, New York, NY, USA).

**Computational docking of Fab ch128.1/IgG1 to hTfR1.** ClusPro docking: The initial coordination of the Fab ch128.1/IgG1 to hTfR1 was performed using the ClusPro docking algorithm[22]. The structures of Fab ch128.1/IgG1 and hTfR1 were uploaded to the ClusPro (version 2.0) server, a rigid body coupling program that uses the PIPER algorithm to find paired interaction potentials and produces a ranking based on the level of stability, interaction energy, and cluster size of models[22,76,77]. In ClusPro 2.0, the structure of Fab ch128.1/IgG1 was defined as "receptor" and the structure of hTfR1 obtained from Protein Data Bank (PDB) deposition 3KAS was defined as "ligand" while ignoring its bound MACV

GP1 subunit of the envelope glycoprotein; structures were docked using "antibody" mode. Given that the amino acid residues S324-S368 of the apical domain of hTfR1 are critical for the binding of ch128.1 antibodies to hTfR1[10], we defined an attraction mask towards these residues using PyMOL. Only surface residues up to a 2.5 Å distance were included in this mask. The docking analysis was ranked by cluster size members in a 9 Å C-alpha RMSD neighborhood to identify the best antibody-antigen coordinates. The best-ranked docking model of the Fab ch128.1/IgG1 to hTfR1 obtained with ClusPro was used as a starting model for redocking with PyRosetta.

PyRosetta docking: Docking was performed with a custom script (Supplementary File: PyRosetta_dockinghfv.py) using PyRosetta software suite (PyRosetta version 3.5, release 84)[23,24]. First, the individual Fab ch128.1/IgG1 and TfR1 were run through PyRosetta's FastRelax function. The individual proteins were then aligned to the output model from ClusPro as a starting position. The structure was then subjected to the docking algorithm that consists of three parts: initial perturbation of Fab ch128.1/IgG1 of 20 Å and a rotation of 10°, low-resolution search in centroid mode and high-resolution refinement. In the low-resolution search phase, the score function 'interchain_cen' was used for minimization. The docking score function was used during high-resolution refinement. The docking protocol was run in parallel through 17,003 individual trajectories to ensure adequate sampling of the conformational space.

**Evaluation of ch128.1/IgG1, ch128.1/IgG1 mutant, and 128.1 treatments in the mouse JUNV challenge model.** For the initial study, 3-week-old mice were sorted by sex prior to infection so that all treatment groups consisted of equivalent percentages of males and females. Mice ($n = 8$/treatment group) were challenged by intraperitoneal (i.p.) injection with $10^{3.5}$ CCID$_{50}$ of JUNV. 1 h after the infection, the animals were treated i.p. with 400 µg of ch128.1/IgG1, ch128.1/IgG1 mutant, or the negative control antibody, IgG1/κ. The mice received a second treatment of 400 µg of the respective antibody on day 3 p.i. This treatment strategy was based on the lethality of the virus in the mouse model and on previous reports using antibodies targeting hTfR1 for cancer therapy in vivo[12,65,66,68]. Sham-infected, untreated controls ($n = 3$) were included to establish weight and clinical score baselines. Animals were weighed daily and observed for clinical signs of disease for 21 days. By 21 days p.i., survivors are generally recovering from the infection, as judged by increasing body weight and the absence of, or limited, clinical disease signs. Clinical disease signs included the presence of weight loss exceeding 10% of peak weight, lethargy, hunched posture, ruffled fur, tremors, paralysis, distended abdomen, and evidence of bleeding from mucosal surfaces. Each of these clinical signs of illness was scored daily ("0" if absent, "1" if present) and animals with a cumulative clinical score greater than 6, or those experiencing weight loss equal to or greater than 30% compared to peak weight or unresponsiveness to external stimulus, were euthanized.

A second experiment was conducted employing a higher challenge dose of JUNV and including a third antibody treatment. Mice (3-week-old) were sorted, as indicated above, prior to i.p. challenge with $10^4$ CCID$_{50}$ of JUNV. After 1 h, the animals were treated i.p. with 400 µg of either ch128.1/IgG1 ($n = 8$), ch128.1/IgG1 mutant ($n = 8$), 128.1 ($n = 6$), IgG1/κ ($n = 8$), or PBS vehicle placebo ($n = 16$). The animals received additional respective antibody treatments of 400 µg each on days 3 and 6 p.i. to attempt to increase the efficacy of the antibody treatments. Sham-infected, untreated controls ($n = 3$) were included to establish weight and clinical score baseline values. Animals were weighed daily and observed for clinical disease signs for 21 days.

**Human colony forming (progenitor) assay.** The colony forming assay was performed as described previously[78,79]. Human BMMC were purchased from STEMCELL™ Technologies (Vancouver, BC, Canada) and plated in quadruplicate according to the manufacturer's instructions. BMMC were seeded in 35 mm dishes (20,000 cells per dish) in MethoCult™ H4434 Classic (a complete methylcellulose medium with recombinant cytokines and erythropoietin; STEMCELL™ Technologies, Cat No. 04434) in the presence of various concentrations of ch128.1/IgG1 ranging from 0.15–75 µg/ml (1–500 nM). Plates were incubated for 14 days at 37 °C in 5% CO$_2$. Negative control treatments consisted of non-treated cells as well as those treated with only the buffer that ch128.1/IgG1 is formulated in (150 mM NaCl, 50 mM Tris, pH 7.8). As a positive control for a toxic compound, BMMC were also treated with 500 nM staurosporine (Thermo Fisher Scientific, Cat. No. BP2541100). Colonies were identified and counted using an Olympus CK2 inverted microscope (Olympus America, Center Valley, PA, USA), and the criteria were defined by STEMCELL™ Technologies for each colony type. Colony types identified were: CFU-E (mature erythroid progenitors), BFU-E (more primitive progenitor than CFU-E), CFU-GM (more mature than CFU-GEMM), and CFU-GEMM (colony forming unit granulocyte/erythroid/macrophage/megakaryocyte). BMMC from three different donors were tested in three independent experiments.

**Erythroid progenitor (erythroblast) differentiation and culture.** Erythroblasts, known to express high levels of hTfR1[28], were differentiated from human bone marrow CD34$^+$ cells (HemaCare, a Charles River Company, Van Nuys, CA, USA) and cultured in StemSpan™ SFEM II medium (Cat. No. 09655) with the addition of StemSpan™ Erythroid Expansion Supplement (100X) (Cat. No. 02692) according to the manufacturer's instructions (STEMCELL™ Technologies). The resulting erythroblast cultures were used in subsequent hTfR1 binding studies.

**hTfR1 binding competition studies.** Erythroblasts ($5 \times 10^5$) were incubated with 5 µg human diferric Tf conjugated to Alexa Fluor™ 546 (Thermo Fisher Scientific, Cat. No. T23364) or 3 µg of recombinant H-Ft conjugated to R-phycoerythrin (H-Ft-PE; Creative Biomart, Shirley, NY, USA, Cat. No. FTH1-528H-PE) for 60 min on ice. Competing antibodies were added to some samples and included 10 µg of either anti-dansyl IgG1/κ isotype control antibody[67], ch128.1/IgG1, or the murine anti-hTfR1 IgG2a/κ antibody clone M-A712 (BD Biosciences, Cat. No. 555534) that is known to block H-Ft binding[33]. Erythroblasts incubated with buffer (PBS) served as a negative unstained control. After the incubation, cells were washed, fixed in 1% paraformaldehyde in PBS and analyzed by flow cytometry using an LSRII analytical flow cytometer and the FACSDiva™ acquisition software (version 8.0.3; BD Biosciences, San Diego, CA, USA). Histograms were created using FCS Express version 3.0 (De Novo Software, Pasadena, CA, USA).

**Antibody-mediated inhibition of Tf and H-Ft internalization.** Tf conjugated to Alexa Fluor™ 546 (5 µg) was added to $5 \times 10^5$ MM.1S human myeloma cells (ATCC® CRL-2974™) in the presence of 10 µg of either the IgG1/κ isotype control or ch128.1/IgG1. All samples were prepared on ice. Some samples were then incubated at 37 °C for 60 min. Other samples were incubated initially on ice for 30 min followed by a 30-min incubation at 37 °C. Non-internalization control samples were incubated on ice for 60 min. After the 60 min incubation, cells were washed, fixed, and analyzed by flow cytometry as described above for the hTfR1 binding competition studies. Similarly, H-Ft-PE (3 µg) was added to $5 \times 10^5$ MM.1S cells in the presence of 10 µg of either the IgG1/κ isotype control, ch128.1/IgG1, or the mouse anti-TfR1 antibody M-A712. Samples were incubated, washed, fixed, and analyzed by flow cytometry as described above for Tf.

The Tf and H-Ft assay samples were also analyzed by imaging flow cytometry using an Amnis ImageStream®X Mk II Imaging Flow Cytometer (Luminex Corporation, Austin, TX, USA) using the INSPIRE™ acquisition software version 201.1.0.744 (Luminex Corporation). Brightfield and fluorescent images were acquired at 60X magnification. During event acquisition, excluding cell debris, 10,000 events were recorded with the following settings: Brightfield LED intensity 32.41 mW; 488 nm laser 200.00 mW. The IDEAS software (Version 6.2; Luminex Corporation) was used for data analysis. Events that were in focus (determined using the Gradient RMS value of 50 or greater on the brightfield channel) and single cells (determined from the brightfield area versus aspect ratio) were selected via gating. A morphology mask of the brightfield image was created using adaptive erode and used to make a mask that excludes the cell membrane and thus reports only internalized fluorescence signal. The define and generate statistics report functions were used to calculate and report the internalized geometric mean fluorescent intensities for Tf conjugated to Alexa Fluor™ 546 and H-Ft-PE. Images were exported using the IDEAS software.

**Statistical analysis.** Survival outcomes in the efficacy studies were analyzed according to the Kaplan and Meier method using the Mantel-Cox log-rank test. Statistical significance in the human colony forming unit assays was determined using the nonparametric, unpaired Mann–Whitney test applying the False Discovery Rate (FDR) approach with two-stage linear step-up method of Benjamini, Krieger, and Yekutieli for multiple comparisons. All statistical evaluations were done using Prism 9 (GraphPad Software, La Jolla, CA, USA). Results were considered significant if $p \leq 0.05$.

**Reporting Summary.** Further information on research design is available in the Nature Research Reporting Summary linked to this article.

## Data availability

The authors declare that the main data supporting the findings of this study are available within the article and its Supplementary Information files. Source data for Figs. 1, 2, 4, 5, and 6 are provided with the paper. Additional data that support the findings of this study have been deposited in the Worldwide Protein Data Bank (wwPDB) with accession code: PDB ID 6WLA and cited in the reference list[72]. All other relevant data are available from the corresponding authors. Source data are provided with this paper.

## Code availability

The custom PyRosetta script, dockinghfv.py, is available in the Zenodo repository (https://doi.org/10.5281/zenodo.5753530) and is cited in the reference list[80].

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

## Acknowledgements

This study was funded in part by the National Institutes of Health (NIH) grant R01CA196266 to Manuel L. Penichet, NIH grant R21AI135851 to Jose A. Rodríguez (UCLA) and by Programa Bec.Ar, Fundación Bunge y Born, Fundación Williams and Fundación René Barón to Gustavo Helguera, who is member of Consejo Nacional de Investigaciones Científicas y Técnicas (CONICET). Portions of this study are based upon research conducted at the Northeastern Collaborative Access Team (NE-CAT) beamlines, funded by the National Institute of General Medical Sciences, NIH (P30 GM124165). The Eiger 16 M detector on the 24-ID-E beamline is funded by an Office of Research Infrastructure Programs (ORIP) High-End Instrumentation (HEI) NIH grant (S10OD021527). Our study used resources of the Advanced Photon Source, a U.S. Department of Energy (DOE) Office of Science User Facility operated by Argonne National Laboratory under Contract No. DE-AC02-06CH11357. We are grateful to Connor Short and Drs. Jose A. Rodríguez, Michael R. Sawaya, and Duilio Cascio (UCLA) for helpful discussions and help in determining the structure of ch128.1/IgG1 Fab using NE-CAT beamline 24-ID-E, funded by NIH-NIGMS P41 GM103403. Figure panels 4a and 5a were created with BioRender.com.

## Author contributions

B.T.H. contributed to conceptualization, design, planning, and completion of experiments, data analysis and interpretation, and writing the manuscript. P.V.C. and T.R.D. contributed to antibody development and characterization, including the toxicity to normal cells and the binding and internalization competition studies, experimental design, data analysis, and manuscript preparation. C.P. contributed to the design, planning, and completion of experiments, docking analysis and structure modeling. L.E.C contributed to the design, planning, completion and interpretation of the bio-layer interferometry competition studies. K.W.B and E.J.S. contributed to planning and completion of experiments. S.Z. and J.Z. contributed to the Fab protein crystallization, X-ray data collection, processing, structure determination and refinement. S.Z performed computational docking in PyRosetta. J.A. contributed to conceptualization, design, planning and interpretation of the bio-layer interferometry competition experiments. G.H. contributed to conceptualization, design, protein crystallization, structure analysis, docking analysis, structure modeling and interpretation and manuscript preparation. M.L.P. contributed to the conceptualization and design of experiments, antibody development, and characterization, including the toxicity to normal cells and the binding and internalization competition studies, data analysis and interpretation and manuscript preparation. B.B.G. contributed to conceptualization, design and planning of experiments, data analysis and interpretation, and writing the manuscript.

## Competing interests

Dr. Manuel Penichet has a financial interest in Stellar Biosciences, Inc. The Regents of the University of California are in discussions with Stellar Biosciences to license a technology invented by Dr. Penichet to this firm. In addition, Dr. Penichet has a financial interest in Klyss Biotech, Inc. All other authors declare no competing interests.

## Ethics approval

All animal procedures complied with USDA guidelines and were conducted at the AAALAC-accredited laboratory animal research facilities at Utah State University under protocol #10034, approved by the Utah State University Institutional Animal Care and Use Committee.
