## [Peer Review File · Nature Communications]

Reviewer #1 (Remarks to the Author):

The manuscript by Hickerson et al. describes the authors' work characterizing an anti-Transferrin Receptor 1 (TfR1) antibody ch128.1, and its derivatives, for their ability to inhibit infection with JUNV in vivo in their recently published hTfR1 transgenic mouse model. Further, the authors provide a crystal structure for their antibody Fab fragments and perform a computational docking to the known structure of hTfR1, which suggests that the binding site may partly overlap with that for MACV GP1, thus providing a theoretical basis for its neutralizing ability.

This paper expands upon previous work characterizing in vitro neutralization by ch128 by providing an important confirmation that this strategy can indeed be viable in vivo, and at the same time provides support for these transgenic mice as a viable model for testing hTfR1 receptor-based therapeutic approaches. Thus, while the experiments are limited and the choice of treatment regimens are not particularly challenging, this clearly represents a significant step in showing the feasibility of this kind of hTfR1-targeting approach for clinical application, which if fully realized would have a significant impact on treatment for these viruses. Further, while the antibody binding data are only based on modelling (rather than co-crystallization) the results would make biological sense in the context of the available data and thus are also informative, although they would be greatly strengthened by the experiment suggested below.

Major Comments:

1) The authors suggest that their antibody acts by physically blocking interaction of TfR1 with the viral glycoprotein, and while this is certainly a feasible explanation, they need to exclude that antibody binding significantly alters TfR1 recycling/surface exposure to further substantiate their position. This would then help to provide much needed biological support for their in silico docking work.

Further, if it would indeed be the case and TfR1 exposure is affected, impairment of iron uptake could in fact be a consideration – something the authors emphasize as a strength of this approach since the antibody does not compete with the binding sites for the major natural ligands of TfR1. However, do the authors think this would really be a problem during an acute treatment course as would presumably be needed in these patients?

2) Lines 276-377. The authors address the issue of possible toxicity associated with antibody treatment, however, they do these experiments in the context of heterologous hTfR1 mice, rather than the homologous mice that provide the background for their experiments. Since these mice still have mTfR1 as well, they would not exclude certain kinds of toxicity (e.g. due to interference with iron uptake) and thus these studies need to be conducted in the appropriate background.

3) To my knowledge, the commonly accepted definition of "host directed therapy" is that it modulates host responses to infection either to achieve antiviral control or limit immunopathology, and what the authors have developed is then correctly an "indirect antiviral therapy". They should be careful not to confuse these different approaches.

Minor Comments:

1) Lines 50-51. While the authors correctly point out that a single mutation (F427I) plays a significant role in attenuation of the Candid#1 vaccine, their statement overlooks that Seregin et al. (REF #6) actually observe that F427I is not fully attenuated and there must be other mutations involved, especially since this mutation only arose after the precursor passages to Candid#1 had already become significantly attenuated. In particular, a possible role for mutations in GP1 has been followed up in that regard (e.g. Manning et al. (2017) Front Cell Infect Microbiol.7:20, Droniou-Bonzom et al. (2011) J Virol. 2011 Dec;85(24):13457-62). But even more importantly, data from the wide-spread use of the vaccine over decades in hundreds of thousands of people demonstrate a solid safety profile. This should not be called into question without clear evidence – indeed doing so could have significant public health consequences if it would turn people off taking

this vaccine, which I am sure is not the authors' intent. Since, the success of the Candid#1 vaccine in no way detracts from the value of the authors work developing alternative (post-exposure) therapies for broad-spectrum treatment of infection with New World arenaviruses they are encouraged to moderate their statements in this regard.

2) Lines 164-166. The authors should briefly clarify their rationale for the selected antibody dose.

3) Lines 231-232. The script for the PyRosetta analysis should be provided as a supplemental file if it has not already been made available on another platform (e.g. Github) – in which case this should be indicated in the text.

4) Lines 319-326. This section does not really appear to fit or to be necessary here (i.e. in the Results section) and would be more appropriate in the Methods.

5) Line 366. The authors should re-define already here what their "mutant antibody" is rather than waiting until line 370 to do this.

6) Discussion. The authors should address whether they anticipate a risk of neurological complications following treatment (i.e. late neurological syndrome) with their hTfR1-directed antibody similar to that reported for JUNV patients treated with convalescent plasma. Or can this in their view be expected to be an additional advantage of a receptor-directed, rather than virus-directed, antibody approach?

7) Figure 2 and 3. For completeness, and to make things easy for the reader, the authors should define their sham-infected groups (grey) in these figures, as they do for the other groups/colours.

8) Line 505. The footprint colour for ch128.1 is the same as for the heavy chain, which is defined as blue (rather than violet) in the rest of the figure. This should be corrected to avoid confusing the reader.

Reviewer #2 (Remarks to the Author):

This manuscript describes the ch128.1/IgG1 antibody directed against human transferrin receptor 1 (hTfR1), which is the cellular receptor used by pathogenic New World mammarenaviruses. The authors demonstrate that this antibody (and a mutant form lacking Fc-function) inhibit Junin virus infection in vitro and can partially protect hTfR1-transgenic mice from lethal Junin challenge. They also solve the crystal structure of the ch128.1/IgG1 Fab and perform in silico docking experiments that predict that the antibody would sterically hinder the virus-receptor interaction.

The work is interesting and the manuscript is well-written, and there is precedent for this type of approach; the approved HIV drug Selzentry (maraviroc) is a small molecule that blocks HIV interaction with its co-receptor CCR5. Unfortunately, though, I do not believe that the significance of the work reported here is sufficient for publication in Nature Communications, especially given that the parental antibody ch128.1/IgG3 was previously published and it looks like the animal model itself is being published elsewhere.

Major points:

- Since this antibody is directed against a host protein, I would be most concerned about any unintended physiologic consequences of administering such an antibody to humans. In their Introduction, the authors suggest that the antibody binding site is away from the regions of hTfR1 that interact with transferrin and with hereditary hemochromatosis protein, but is known to interact with ferritin. Despite not seeing significant adverse effects in their sham-infected transgenic mouse model, I still think that the authors need to demonstrate that the ch128.1/IgG1 antibody has no effect on the normal known functions of hTfR1 before this host-targeted strategy can really be proposed for therapeutic development.

- Fig. 3: The ch128.1/IgG1 mutant treated mice, while partially protected from lethal infection (Fig. 3B) and most clinical disease (Fig. 3D), did not exhibit significant weight gains like the sham-infected or the ch128.1/IgG1 group. Or indeed in the group receiving the same antibody in the

previous experiment (Fig. 2C). What might explain this observation?

- The authors perform in silico docking to predict that ch128.1/IgG1 and Machupo GP1 bind to an overlapping region on hTfR1 and suggest that the blocking activity is through steric hindrance. While such predictions are useful, I would suggest that for a high-profile journal such as Nature Communications, these should be accompanied by biological data. Does ch128.1/IgG1 compete with recombinant GP1 binding to hTfR1? Does it compete with labelled virus binding to hTfR1?

Minor points:

Line 248: How do the EC90 values with the ch128.1/IgG1 antibody compare with the ch128.1/IgG3? Is there any loss of activity associated with switching isotype? It might be worth pointing this out here, to save the reader from going back to look at the previous paper.

Reviewer #3 (Remarks to the Author):

The authors address a very important medical need in this study: the development of a broad-spectrum antiviral for treatment of NW arenavirus infection. They bring a highly novel model of JUNV infection and disease and a strong candidate antiviral antibody targeting hTfR1 to the study. Overall, this is a well-written paper and a well-controlled and rigorous study. The results certainly support the overall conclusions. The novelty of the study is somewhat dampened by the fact that both the antibody and mouse model have been published previously. However, showing that an hTfR1-directed antiviral can protect in vivo is novel and not only suggests the current antibody is worthy of further consideration, but it also opens the door for other leading candidates that also target hTfR1.

As the study currently stands, the authors clearly show the benefit of antibody treatment if the antibody is used in a completely prophylactic mode. However, many (perhaps most) instances where this antibody would be used would be in symptomatic individuals. I think the paper would benefit greatly from another challenge round where mice receive treatment at the time of symptom onset. This would be a logical test of the utility of this antibody in real-world settings of medical need.

And a very minor comment relates to lines 71-72. I think it is well accepted that the suspected etiological role of WWAV in those fatalities was due to a PCR contamination in the diagnostic lab. Maybe just leave it at the fact that WWAV is susceptible to the antibody. No need to perpetuate that erroneous result, despite the existing refs.

Overall, strong work by this talented team and it was a pleasure to review. And my apologies to the authors for the slow timeline for my review. COVID-19 seems to infiltrate every facet of virology research these days.

Response to reviewer comments

Reviewer #1 (Remarks to the Author):

The manuscript by Hickerson et al. describes the authors' work characterizing an anti-Transferrin Receptor 1 (TfR1) antibody ch128.1, and its derivatives, for their ability to inhibit infection with JUNV in vivo in their recently published hTfR1 transgenic mouse model. Further, the authors provide a crystal structure for their antibody Fab fragments and perform a computational docking to the known structure of hTfR1, which suggests that the binding site may partly overlap with that for MACV GP1, thus providing a theoretical basis for its neutralizing ability.

This paper expands upon previous work characterizing in vitro neutralization by ch128 by providing an important confirmation that this strategy can indeed be viable in vivo, and at the same time provides support for these transgenic mice as a viable model for testing hTfR1 receptor-based therapeutic approaches. Thus, while the experiments are limited and the choice of treatment regimens are not particularly challenging, this clearly represents a significant step in showing the feasibility of this kind of hTfR1-targeting approach for clinical application, which if fully realized would have a significant impact on treatment for these viruses. Further, while the antibody binding data are only based on modelling (rather than co-crystallization) the results would make biological sense in the context of the available data and thus are also informative, although they would be greatly strengthened by the experiment suggested below.

Major Comments:

1) The authors suggest that their antibody acts by physically blocking interaction of TfR1 with the viral glycoprotein, and while this is certainly a feasible explanation, they need to exclude that antibody binding significantly alters TfR1 recycling/surface exposure to further substantiate their position. This would then help to provide much needed biological support for their in silico docking work.

We have now performed new experiments using a bio-layer interferometry competition assay (see new Figure 2) that demonstrate that ch128.1 and MACV GP1 bind to an overlapping region of hTfR1. The competition assay employed a novel MACV GP1-Fc fusion protein containing the Fc region of human IgG1 (reported in our manuscript for the first time), the ch128.1 Fab and soluble human transferrin receptor 1 (sTfR1). In addition, we now also reference previous work evaluating competitive binding by ch128.1/IgG3 and a MACV GP1-Fc fusion protein (containing the Fc region of murine IgG2a) to human cells expressing TfR1 by flow cytometry. The new and previously reported data are included and discussed in the revised manuscript (2nd section of the Results and the last sentence of the 4th paragraph of the Discussion) providing mechanistic detail in support of the molecular modelling data.

We also performed transferrin binding (see new Figure 7) and internalization (see new Figure 8) experiments and definitively show that these functions are not affected by treatment with ch128.1/IgG1 (see new Figures 7 and 8). The new data are included and discussed in the revised manuscript (last section of the Results and the last sentence of the 2nd to last paragraph of the Discussion).

Further, if it would indeed be the case and TfR1 exposure is affected, impairment of iron uptake could in fact be a consideration – something the authors emphasize as a strength of this approach since the antibody does not compete with the binding sites for the major natural

ligands of TfR1. However, do the authors think this would really be a problem during an acute treatment course as would presumably be needed in these patients?

This is a very relevant point. However, during the course of short-term treatment, we would not anticipate that impairment of iron uptake would be a major problem. Also, as stated above, we have shown that ch128.1 is a non-neutralizing antibody, meaning that it does not inhibit the binding of the major natural ligand of TfR1 responsible for iron uptake (transferrin), and transferrin internalization is unaffected by treatment with the antibody (new Figures 7 and 8). In addition, ch128.1 only partially inhibited the uptake of H-Ft (new Figure 9).

It is important to stress that several anti-TfR1 antibodies, including antibodies that inhibit the binding of transferrin, have been shown to be well-tolerated in general and only result in transient anemia in nonhuman primates, supporting their clinical evaluation in cancer patients. An example is the human IgG1 JST-TFR09 (also known as PPMX-T003) that shows only mild anemia in a few cases in cynomolgus macaques (*Macaca fascicularis*) as the only toxicity, even with repeated administration of the antibody at high doses of 30 mg/kg (Shimosaki et al., *Biochem Biophys Res Commun* (2017) 485: 144-151; Zhang et al., *AACR Annual Meeting*, (2017) Abstract No. 5586). These findings have supported a recently initiated clinical trial for cancer therapy (late 2019, <https://www.ppmx.com/en/rd/pipeline.html>). Another example is the Phase I clinical trial of the murine IgA 42/6 antibody, in which the therapy was generally well-tolerated in cancer patients, even at high doses (Brooks et al., *Clin Cancer Res* (1995) 1: 1259-1265) and the murine IgG2b A24 antibody that caused a slight decrease of hemoglobin levels in cynomolgus monkeys (Moura et al., *Retrovirology* (2011) 8 (Suppl 1): A60). Moreover, the non-neutralizing R17 217 IgG2a rat anti-mouse TfR1 showed no toxicity in mice, with toxicity only observed when the antibody was conjugated with a potent toxin (Bjorn and Groetsema, *Cancer Res* (1987) 47: 6639-6645).

Based on the aforementioned results, non-neutralizing antibodies of the IgG class, such as ch128.1, that do not interfere with transferrin binding are also expected to be at least as well-tolerated compared to their neutralizing counterparts. It is also important to clarify that in the unlikely event of acute anemia, the condition would most likely be temporary and can be treated by transfusion and/or iron supplements. Also of note, hematopoietic stem cells would not be affected by the ch128.1 since these cells express little to no TfR1.

2) Lines 276-377. The authors address the issue of possible toxicity associated with antibody treatment; however, they do these experiments in the context of heterologous hTfR1 mice, rather than the homologous mice that provide the background for their experiments. Since these mice still have mTfR1 as well, they would not exclude certain kinds of toxicity (e.g. due to interference with iron uptake) and thus these studies need to be conducted in the appropriate background.

Although this is a valid point, we are in the early stages of development. Thus, our strategy was not to use the present studies to validate the antibodies for clinical use, but to demonstrate proof-of-concept that our approach is viable *in vivo*. In addition, the hTfR1 mouse model used in our report is not ideal for toxicity studies since the pattern of hTfR1 expression is not physiological, as mentioned in Ref. 21 (Yu et al., *Sci Transl Med* (2014) 6: 261ra154), and thus, unlikely to yield a meaningful outcome that can be translated into humans.

In-depth toxicology will be pursued in more sophisticated rodent models currently in development, and ultimately in a nonhuman primate model that would provide more meaningful results. These studies, however, are outside of the scope of the present report.

In addition, we face limitations in the current MTA with Genentech, Inc. (the source of the hTfR1 mouse) that make it difficult to conduct additional animal studies using the JUNV hTfR1 mice. However, as mentioned above, we are working on the development of more sophisticated rodent models that will be used in future toxicological assessments.

3) To my knowledge, the commonly accepted definition of “host directed therapy” is that it modulates host responses to infection either to achieve antiviral control or limit immunopathology, and what the authors have developed is then correctly an “indirect antiviral therapy”. They should be careful not to confuse these different approaches.

We understand the point raised by the reviewer. To address this concern, we have changed “host-directed” to “host receptor-targeted” throughout the revised manuscript.

Minor Comments:

1) Lines 50-51. While the authors correctly point out that a single mutation (F427I) plays a significant role in attenuation of the Candid#1 vaccine, their statement overlooks that Seregin et al. (REF #6) actually observe that F427I is not fully attenuated and there must be other mutations involved, especially since this mutation only arose after the precursor passages to Candid#1 had already become significantly attenuated. In particular, a possible role for mutations in GP1 has been followed up in that regard (e.g. Manning et al. (2017) Front Cell Infect Microbiol.7:20, Droniou-Bonzom et al. (2011) J Virol. 2011 Dec;85(24):13457-62). But even more importantly, data from the wide-spread use of the vaccine over decades in hundreds of thousands of people demonstrate a solid safety profile. This should not be called into question without clear evidence – indeed doing so could have significant public health consequences if it would turn people off taking this vaccine, which I am sure is not the authors’ intent. Since, the success of the Candid#1 vaccine in no way detracts from the value of the authors work developing alternative (post-exposure) therapies for broad-spectrum treatment of infection with New World arenaviruses they are encouraged to moderate their statements in this regard.

Indeed, our intent is not to discourage vaccination with Candid#1 in any way. As such, we have removed the following statement from the revised manuscript.

“However, recent evidence of reversion to virulence through a single mutation has raised concerns regarding the safety of the Candid#1 vaccine⁵⁻⁷.”

2) Lines 164-166. The authors should briefly clarify their rationale for the selected antibody dose.

In our studies, we used a 400 µg dose followed by additional 400 µg doses on day 3 (both experiments) and day 6 (2nd experiment only). The day 6 treatment was included in the 2nd experiment to attempt to increase the efficacy of the antibody treatments. Since this is the first time that an antibody targeting TfR1 was being used to protect against mammarenavirus infection *in vivo*, we could not use previous reports to support a dose and schedule rationale. However, this treatment strategy was consistent with multiple publications using anti-TfR1 antibodies for cancer therapy *in vivo*. Based on the reviewer’s comment, we have clarified this point in the revised manuscript (page 7, 3rd paragraph, 5th sentence) stating that our treatment strategy was based on the lethality of the virus in the mouse model and on previous reports using antibodies targeting TfR1 for cancer therapy *in vivo*.

3) Lines 231-232. The script for the PyRosetta analysis should be provided as a supplemental file if it has not already been made available on another platform (e.g. Github) – in which case this should be indicated in the text.

The python script that we used for docking with PyRosetta is now provided as the supplemental file “PyRosetta_docking_hfv.py” and this is now indicated in the Materials and Methods section of the revised manuscript.

4) Lines 319-326. This section does not really appear to fit or to be necessary here (i.e. in the Results section) and would be more appropriate in the Methods.

We agree with the reviewers point. Because most of the text in this section was already explained in the Materials and Methods section, much was deleted with limited text to provide clarity for the analysis performed and brief description of docking funnel, which is the key result.

5) Line 366. The authors should re-define already here what their “mutant antibody” is rather than waiting until line 370 to do this.

We have adopted the reviewer’s recommendation and now define the impairment in Fc receptor binding at this earlier position in the text. This change can be seen in the Discussion, 2nd paragraph, 4th to last sentence of the revised manuscript.

6) Discussion. The authors should address whether they anticipate a risk of neurological complications following treatment (i.e. late neurological syndrome) with their hTfR1-directed antibody similar to that reported for JUNV patients treated with convalescent plasma. Or can this in their view be expected to be an additional advantage of a receptor-directed, rather than virus-directed, antibody approach?

We have added text, at the end of the 2nd paragraph of the Discussion, to make the point that additional studies are required to assess whether the hTfR1 model recapitulates the late neurology syndrome (LNS) which is observed in a proportion of surviving patients treated with immune plasma and whether ch128.1 would reduce the risk of relapse to LNS. We would rather not speculate on the latter without supporting evidence.

7) Figure 2 and 3. For completeness, and to make things easy for the reader, the authors should define their sham-infected groups (grey) in these figures, as they do for the other groups/colours.

The sham-infected groups are defined in the legend as a gray line with an asterisk symbol just above the summary tables in what are now Figures 4 and 5 in the revised manuscript. Because the sham-infected control mice do not receive treatment (untreated), they are not included in the A panels where the color-coded treatments are listed.

8) Line 505. The footprint colour for ch128.1 is the same as for the heavy chain, which is defined as blue (rather than violet) in the rest of the figure. This should be corrected to avoid confusing the reader.

We thank the reviewer for bringing this matter to our attention and have made the following improvements to the figure.

1. New coloring of the heavy chain and light chain in panel A, and the change in color of the CDR-L to orange.
2. New coloring in panel C to match the heavy chain (blue) and light chain (light blue) of the constant regions shown in panel A.
3. New coloring in panel D to show in blue the footprint of the heavy chain, in light blue the footprint of the light chain, in pink the footprint of MACV-GP1 and in red the overlapping residues.

Reviewer #2 (Remarks to the Author):

This manuscript describes the ch128.1/IgG1 antibody directed against human transferrin receptor 1 (hTfR1), which is the cellular receptor used by pathogenic New World mammarenaviruses. The authors demonstrate that this antibody (and a mutant form lacking Fc-function) inhibit Junin virus infection *in vitro* and can partially protect hTfR1-transgenic mice from lethal Junin challenge. They also solve the crystal structure of the ch128.1/IgG1 Fab and perform *in silico* docking experiments that predict that the antibody would sterically hinder the virus-receptor interaction.

The work is interesting and the manuscript is well-written, and there is precedent for this type of approach; the approved HIV drug Selzentry (maraviroc) is a small molecule that blocks HIV interaction with its co-receptor CCR5. Unfortunately, though, I do not believe that the significance of the work reported here is sufficient for publication in Nature Communications, especially given that the parental antibody ch128.1/IgG3 was previously published and it looks like the animal model itself is being published elsewhere.

Major points:

- Since this antibody is directed against a host protein, I would be most concerned about any unintended physiologic consequences of administering such an antibody to humans. In their Introduction, the authors suggest that the antibody binding site is away from the regions of hTfR1 that interact with transferrin and with hereditary hemochromatosis protein, but is known to interact with ferritin. Despite not seeing significant adverse effects in their sham-infected transgenic mouse model, I still think that the authors need to demonstrate that the ch128.1/IgG1 antibody has no effect on the normal known functions of hTfR1 before this host-targeted strategy can really be proposed for therapeutic development.

Potential adverse effects associated with the host TfR1-targeted strategy described is a valid concern. To address this concern, we conducted additional studies to: 1) provide experimental evidence that our anti-hTfR1 antibody does not significantly interfere with transferrin or ferritin uptake and 2) include data assessing *in vitro* cytotoxicity. We now include standard flow cytometry, as well as imaging flow cytometry, data confirming that the ch128.1/IgG1 antibody does not compete with transferrin for binding to hTfR1 (new Figure 7) and transferrin uptake (new Figure 8) is not affected by the antibody. The ch128.1/IgG1 antibody, however, does weakly compete with ferritin binding (new Figure 7) and uptake (new Figure 9) compared to a positive control antibody (M-A712) that is well known to inhibit the binding of ferritin to hTfR1. This result is consistent with the known binding of ch128.1/IgG1 to a region on the apical domain of hTfR1 that partially overlaps with the ferritin-binding domain. We also demonstrate using a widely-accepted colony forming assay that evaluated toxicity to committed human hematopoietic progenitor cell assays (CFU-E, BFU-E, CFU-GM and CFU-GEMM) that ch128.1/IgG1 has low cytotoxicity. Importantly, these assays are commonly used to study the effect of anti-cancer agents, including anti-TfR1 antibodies, on hematopoiesis *in vitro*.

- Fig. 3: The ch128.1/IgG1 mutant treated mice, while partially protected from lethal infection (Fig. 3B) and most clinical disease (Fig. 3D), did not exhibit significant weight gains like the sham-infected or the ch128.1/IgG1 group. Or indeed in the group receiving the same antibody in the previous experiment (Fig. 2C). What might explain this observation?

It is common to see some variability in the weight change parameter, especially with young mice, due to inherent experiment-to-experiment variability in mouse efficacy studies. Also, a slightly higher JUNV challenge dose was used in the second experiment and the mice received three antibody treatments as opposed to two in the first experiment.

- The authors perform in silico docking to predict that ch128.1/IgG1 and Machupo GP1 bind to an overlapping region on hTfR1 and suggest that the blocking activity is through steric hinderance. While such predictions are useful, I would suggest that for a high-profile journal such as Nature Communications, these should be accompanied by biological data. Does ch128.1/IgG1 compete with recombinant GP1 binding to hTfR1? Does it compete with labelled virus binding to hTfR1?

We have now performed competition experiments (see new Figure 2) that demonstrate that ch128.1 and MACV GP1 bind to an overlapping region of hTfR1. In addition, we now also reference previous work evaluating competitive binding by ch128.1/IgG3 and a MACV GP1-Fc fusion protein to human cells expressing TfR1 by flow cytometry (Helguera et al., *J Virol* (2012) 86: 4024-4028). The new and previously reported data are included and discussed in the revised manuscript (2nd section of the Results and the last sentence of the 4th paragraph of the Discussion) providing mechanistic detail in support of the molecular modelling data.

Minor points:

Line 248: How do the EC90 values with the ch128.1/IgG1 antibody compare with the ch128.1/IgG3? Is there any loss of activity associated with switching isotype? It might be worth pointing this out here, to save the reader from going back to look at the previous paper.

In the previous work by Helguera et al., the EC50 values using the ch128.1/IgG3 were determined for arenavirus pseudotyped viruses only, not infectious JUNV. With the attenuated JUNV strain IV445, the tested dose of 200 nM of the ch128.1/IgG3 strongly inhibited the infection, but EC50 and EC90 concentrations were not determined, and therefore a direct comparison cannot be made. We have revised the text in the first paragraph of the Results section to provide this additional context for the reader.

Reviewer #3 (Remarks to the Author):

The authors address a very important medical need in this study: the development of a broad-spectrum antiviral for treatment of NW arenavirus infection. They bring a highly novel model of JUNV infection and disease and a strong candidate antiviral antibody targeting hTfR1 to the study. Overall, this is a well written paper and a well-controlled and rigorous study. The results certainly support the overall conclusions. The novelty of the study is somewhat dampened by the fact that both the antibody and mouse model have been published previously. However, showing that an hTfR1 directed antiviral can protect in vivo is novel and not only suggests the current antibody is worthy of further consideration, but it also opens the door for other leading

candidates that also target hTfR1.

As the study currently stands, the authors clearly show the benefit of antibody treatment if the antibody is used in a completely prophylactic mode. However, many (perhaps most) instances where this antibody would be used would be in symptomatic individuals. I think the paper would benefit greatly from another challenge round where by mice receive treatment at the time of symptom onset. This would be a logical test of the utility of this antibody in real world settings of medical need.

Our main goal was to demonstrate *in vivo* proof-of-concept. Therapeutic efficacy will be pursued in more sophisticated rodent models currently in development, and ultimately a nonhuman primate model that would provide more meaningful results.

And a very minor comment relates to lines 71-72. I think it is well accepted that the suspected etiological role of WWAV in those fatalities was due to a PCR contamination in the diagnostic lab. Maybe just leave it at the fact that WWAV is susceptible to the antibody. No need to perpetuate that erroneous result, despite the existing refs.

The text has been modified as suggested (see tracked deletion, 3rd paragraph of Introduction).

Overall, strong work by this talented team and it was a pleasure to review. And my apologies to the authors for the slow timeline for my review. COVID-19 seems to infiltrate every facet of virology research these days.

We are grateful to all the reviewers for their meaningful comments and time invested in such careful review of our manuscript, especially during these difficult times of the COVID-19 pandemic.

Peer review comments, further review

Reviewer #1 (Remarks to the Author):

The authors have done a thorough job of addressing my comments. In particular, they have provided a significant amount of additional data substantiating the proposed mechanism of action for their antibody treatment (and excluding effects on other biological activities of TfR1).

While there were a few points that could not be directly addressed, the authors' responses are reasonable and this does not take away from the fact that, taken together, this is a solid piece of research and makes a major contribution to moving in vivo research on TfR1-directed therapies forward.

I have no further suggestions for the authors.

Reviewer #2 (Remarks to the Author):

In this new version of the manuscript the authors have successfully addressed all my previous concerns.

Reviewer #3 (Remarks to the Author):

While I still think a therapeutic challenge study would strengthen the study, I can agree with the authors that they have indeed provided proof-of-concept for this mAb and the concept that hTfR1 targeting is worth pursuing. I will therefore wait to see the results of therapeutic studies in a follow up manuscript. And I applaud their efforts to address the other reviewers' comments. I think the new work greatly strengthens the MS.

Response to reviewers' comments

REVIEWERS' COMMENTS

Reviewer #1 (Remarks to the Author):

The authors have done a thorough job of addressing my comments. In particular, they have provided a significant amount of additional data substantiating the proposed mechanism of action for their antibody treatment (and excluding effects on other biological activities of TfR1).

While there were a few points that could not be directly addressed, the authors' responses are reasonable and this does not take away from the fact that, taken together, this is a solid piece of research and makes a major contribution to moving in vivo research on TfR1-directed therapies forward.

I have no further suggestions for the authors.

Reviewer #2 (Remarks to the Author):

In this new version of the manuscript the authors have successfully addressed all my previous concerns.

Reviewer #3 (Remarks to the Author):

While I still think a therapeutic challenge study would strengthen the study, I can agree with the authors that they have indeed provided proof-of-concept for this mAb and the concept that hTfR1 targeting is worth pursuing. I will therefore wait to see the results of therapeutic studies in a follow up manuscript. And I applaud their efforts to address the other reviewers' comments. I think the new work greatly strengthens the MS.

We are pleased to have addressed all major comments and concerns satisfactorily and thank the reviewers for their time and efforts.